IFT-UAM/CSIC-23-48

# Electric conductivity in non-Hermitian holography

Zhuo-Yu Xian[1,2], David Rodríguez Fernández[3-5,*], Zhaohui Chen[1,2], Yang Liu[1], René Meyer[1,2]

**1** Institute for Theoretical Physics and Astrophysics, Julius–Maximilians–Universität Würzburg, Am Hubland, 97074 Würzburg, Germany
**2** Würzburg-Dresden Cluster of Excellence ct.qmat
**3** Instituut-Lorentz for Theoretical Physics, Universiteit Leiden, P. O. Box 9506, 2300 RA Leiden, The Netherlands
**4** Instituto de Física Teórica UAM/CSIC, Calle Nicolás Cabrera 13-15, 28049 Madrid, Spain
**5** Departamento de Física Teórica, Universidad Autónoma de Madrid, Campus de Cantoblanco, 28049 Madrid, Spain
*Corresponding author: david.rodriguez01@uam.es

## Abstract

We study the phase structure and charge transport at finite temperature and chemical potential in the non-Hermitian $\mathcal{PT}$-symmetric holographic model of [1]. The non-Hermitian $\mathcal{PT}$-symmetric deformation is realized by promoting the parameter of a global $U(1)$ symmetry to a complex number. Depending on the strength of the deformation, we find three phases: stable $\mathcal{PT}$-symmetric phase, unstable $\mathcal{PT}$-symmetric phase, and an unstable $\mathcal{PT}$-symmetry broken phase. In the three phases, the square of the condensate and also the spectral weight of the AC conductivity at zero frequency are, respectively, positive, negative, and complex. We check that the Ferrell-Glover-Tinkham sum rule for the AC conductivity holds in all the three phases. We also investigate a complexified $U(1)$ rotor model with $\mathcal{PT}$-symmetric deformation, derive its phase structure and condensation pattern, and find a zero frequency spectral weight analogous to the holographic model.

# 1  Introduction

Non-Hermitian Hamiltonian evolution has attracted increasing interest in various areas of physics, including condensed matter, quantum information, and AdS/CFT, for reviews c.f. e.g. [2, 3]. In condensed matter, it has been widely employed in describing open quantum systems [4], for example as effective models of the finite quasiparticle lifetime introduced by electron-electron or electron-phonon interactions [5]. Furthermore, non-Hermitian descriptions have been employed in Weyl semimetals [6–8], for delocalization transitions or vortex flux line depinning in type II superconductors [9], as well as in the context of the interplay between topology and dissipation [10,11]. In quantum information, non-Hermiticity has been introduced to describe projective measurements on quantum circuits or many-body systems, which turns out to be an efficient way to prepare entangled states [12–15] and conduct quantum teleportation [16]. In AdS/CFT, it is also a potential route to a better understanding of the holography of complex spacetime metrics and of quantum matter [17–21].

Non-Hermiticity does not necessarily lead to a complex energy spectrum and non-unitary Hamiltonian evolution [3, 22, 23]. If a non-Hermitian Hamiltonian satisfies $\mathcal{PT}$-symmetry, namely the Hamiltonian is invariant under the combination of a generalized time-reversal $\mathcal{T}$ and a generalized parity $\mathcal{P}$ transformation, it is possible that the spectrum remains real and unitary evolution still holds in terms of a new inner product. $\mathcal{PT}$-symmetric theories have been extensively explored in the context of quantum mechanics [22,23], quantum field theory [3, 24–26], and even classical physics [27]. If an eigenstate of the Hamiltonian is also a simultaneous eigenstate of the operator $\mathcal{PT}$, then its energy is real. If it is not a $\mathcal{PT}$ eigenstate but part of a $\mathcal{PT}$ doublet, it is in general mapped to another state by the action of $\mathcal{PT}$, with complex conjugate energy. Most $\mathcal{PT}$-symmetric Hamiltonians are found to be pseudo-Hermitian, with eigenenergies appearing in complex conjugate pairs [28,29]. The spectrum of a $\mathcal{PT}$-symmetric Hamiltonian can hence be real, partially complex, or completely complex. If at least some energies are complex, the $\mathcal{PT}$ symmetry is spontaneously broken. A $\mathcal{PT}$-symmetric Hamiltonian with a real spectrum can always be related to a Hermitian Hamiltonian via a similarity transformation [28,30–32], the so-called Dyson map [33]. Recently, it has been observed that $\mathcal{PT}$ symmetry also plays an increasingly important role in strongly interacting systems relevant to holography, such as the Sachdev-Ye-Kitaev model [34–37] and holographic quantum matter [1,38].

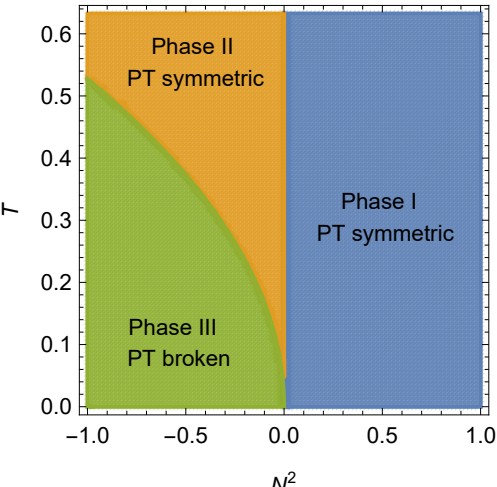

| | Phase I | Phase II | Phase III |
|---|---|---|---|
| $\mathcal{PT}$ symmetry | preserved | preserved | broken |
| (free) energy | real | real | complex |
| scalar stability | stable | unstable | unstable |
| vector stability | stable | stable | stable |
| superfluid density | positive | negative | complex |
| QC conductivity | suppressed | enhanced | complex |
| FGT sum rule | holds | holds | holds |
| NEC | holds | violated | ill-defined |

Figure 1: Phase diagram of the $\mathcal{PT}$-symmetric model (4), with parameters $d = 3$, $m^2 = -2$, $q = 1$ and $v = 3/2$, at zero chemical potential $\mu = 0$.

Table 1: The properties of the three phases of the model of [1].

In this work, we will focus on the electric conductivity of the $\mathcal{PT}$-symmetric non-Hermitian model of [1]. Unitarity constrains transport phenomena in electron systems as well as in holography, for example by constraining the shear and bulk viscosities $\eta$ and $\zeta$, as well as the dissipative part of the electric conductivity $\mathrm{Re}\,\sigma$ to be positive semidefinite [39],

$$\eta \geq 0\,, \quad \zeta \geq 0\,, \quad \mathrm{Re}\,\sigma \geq 0\,. \tag{1}$$

Investigating transport in strongly interacting non-Hermitian systems via the AdS/CFT correspondence is of interest both to high energy and to condensed matter. We expect that the AdS/CFT correspondence will provide predictions for transport coefficients in strongly coupled and correlated $\mathcal{PT}$-symmetric systems, which are unamenable to perturbation theory otherwise.

In the model of [1], the non-Hermitian deformation is implemented by complexifying the global $U(1)$ symmetry that is spontaneously broken in the Einstein-Maxwell-scalar model of [40]. As reviewed in Sec. 2.1, this effectively decouples the source for the charged scalar operator from its complex conjugate. This implements the Dyson map which connects a $\mathcal{PT}$-symmetric Hamiltonian with real spectrum with an ordinary Hermitian Hamiltonian. We first investigate the phase structure in the presence of the non-Hermitian $\mathcal{PT}$-symmetric deformation at finite temperature and both at zero and finite chemical potential. The phase diagram at zero chemical potential is shown in Fig. 1 and discussed in Sec. 2.3.2. At zero chemical potential, there are three finite temperature phases in the model of [1]: We not only reproduce the real solution in phase I and the pair of purely imaginary solutions in phase II found in [1], but also numerically construct a pair of complex conjugate solutions in phase III. Thee two complex conjugate solutions admit complex temperatures, and we check that their zero temperature limit approaches the two zero temperature solutions already found in [1]. The phase diagram at finite chemical potential, discussed in Sec. 2.3.2 and shown in Fig. 4, is similar to the zero chemical potential case. We will comment on the interplay of superconductivity and $\mathcal{PT}$ breaking in Sec. 5.

The main results of our work are the calculation of the AC electric conductivity from linear perturbation theory, and the verification of the Ferrell-Glover-Tinkham (FGT) sum

rule in each phase. The AC conductivity itself shows an interesting structure in phases I, II and III: In phase I the expectation value of the operator induces a positive superfluid density $\rho_s$ that leads to a $\rho_s\delta(\omega)$ contribution to $\mathrm{Re}\,\sigma_{xx}$ and an associated $1/\omega$ pole in $\mathrm{Im}\,\sigma_{xx}$. In phase II, the superfluid density turns negative, $\rho_s < 0$. In phase III, $\rho_s$ itself becomes complex, leading to a $\delta(\omega)$ contribution and a $1/\omega$ pole in both $\mathrm{Re}\,\sigma_{xx}$ and $\mathrm{Im}\,\sigma_{xx}$. In general, the AC conductivity in holographic systems consist of three different parts, describing the effects of quantum criticality (incoherent transport), superconductivity, and coherent transport. They all contribute additively to the low frequency conductivity [41–49],

$$\sigma(\omega) = \sigma_i(\omega) + \sigma_s(\omega) + \sigma_c(\omega)\,. \tag{2}$$

The incoherent conductivity $\sigma_i(\omega)$ is defined as the part of the conductivity unrelated to either momentum transport or condensation. Its $\omega \to 0$ limit is usually called the quantum critical conductivity, $\sigma_Q = \sigma_i(0)$. In the weakly coupled limit, $\sigma_Q$ (and actually $\sigma_i(\omega)$) originates from the momentum-conserving scattering between electrons and holes [50] and is thus incoherent with the momentum flow, hence the name. The superconducting contribution $\sigma_s(\omega) = \rho_s q^2 \left(\pi\delta(\omega) + i/\omega\right)$ is induced by the condensation of normal state charge carriers [51–54], where $q$ is the charge of Cooper pairs. The coherent conductivity $\sigma_c(\omega)$ originates from the charge flow that is coherent to the momentum flow. In the absence of momentum relaxation, it has a $\pi\delta(\omega)+i/\omega$ contribution analogous to $\sigma_s$, as well as an analytic part. As electric charge is conserved, the charge carriers could contribute to all three parts of the conductivity by spectral weight transfer. As evident from Figs. 6, 7 and 8, we find that $\sigma_Q$ is reduced in phase I as compared to the AdS-Schwarzschild value $\sigma_Q = 1$, enhanced in phase II, and becomes complex in phase III. In Sec. 4, we study a complexified $U(1)$ rotor model with the same $\mathcal{PT}$-symmetric deformation as in [1], which reproduces the phases of the holographic model, and whose zero frequency spectral weight coincides with our results for the holographic model of [1].

Finally, we checked the validity of the FGT sum rule in all three phases. In $d = 3$ $(2+1$ boundary dimensions), at high frequencies the AC electric conductivity calculated in an asymptotically $\mathrm{AdS}_4$ background tends to a constant $\mathrm{Re}\,\sigma_{xx} = 1$ (in units of $e^2/h$), due to the scale invariance of the ultraviolet (UV) fixed point. Following [55], we subtract this constant, after which the FGT sum rule reads

$$\int_{-\infty}^{\infty} \mathrm{Re}\left[\sigma(\omega) - 1\right] d\omega = 0\,. \tag{3}$$

We take the integral in (3) over the whole real axis, as the symmetry of conductivity under the transformation $\omega \leftrightarrow -\omega$ may not hold in the presence of non-Hermiticity. The subtraction is necessary for the integral to converge. The sum rule is expected to hold under the assumptions of causality, unitarity, and charge conservation [55]. It has been verified in various holography models [46, 55–59] fulfilling these assumptions. Since $\mathcal{PT}$-symmetric non-Hermitian systems can break some of these assumptions, in particular unitarity in the $\mathcal{PT}$-broken phase, we check the validity of (3) in the holographic model of [1]. We find that the sum rule holds in all three phases, even in the $\mathcal{PT}$-broken phase.

This paper is organized as follows. In Sec. 2, we introduce the Einstein-Maxwell-scalar theory with the non-Hermitian $\mathcal{PT}$-symmetric source deformation of [1], and study its phase structure at finite temperature and chemical potential. In Sec. 3, we calculate and discuss the AC conductivity. In addition, we verify the sum rule in each phase. In Sec. 4, we study a complexified $U(1)$ rotor model with a $\mathcal{PT}$-symmetric deformation. We find that its phase structure and zero frequency spectral weight turn out analogous to our holographic model. In Sec. 5, we present our conclusions and outline further research directions.

# 2 Non-Hermitian Holography

In this section, we first introduce the $\mathcal{PT}$-symmetric non-Hermitian model of [1], and review its symmetries and the associated Dyson map, in Sec. 2.1. In Sec. 2.2, we present the equations of motion and the ansatz for the bulk fields that we will solve numerically in the remainder of this section for both vanishing (Sec. 2.3) and finite (Sec. 2.4) chemical potential.

## 2.1 Action and symmetries

In the AdS/CFT correspondence, every continuous global symmetry of the dual field theory is represented by a gauge symmetry in the bulk. Thus, while constructing the bulk theory, we at least need a $U(1)$ gauge symmetry whose gauge field encodes the conserved current on the boundary. In addition, in order to implement the $\mathcal{PT}$ deformation of [1], a charged bulk field is needed. We follow the previous works [1,38] and consider the scalar field to be minimally coupled, with charge $q$ under the $U(1)$ symmetry. The holographic action reads

$$S = \int d^{d+1}x\sqrt{-g}\left[R + \frac{d(d-1)}{L^2} - D_a^\dagger\bar{\phi}D^a\phi - m^2\bar{\phi}\phi - v\bar{\phi}^2\phi^2 - \frac{1}{4}F_{ab}F^{ab}\right], \quad (4)$$

with $D_a = \nabla_a - iqA_a$ and $16\pi G_N = 1$. In this paper we consider $d = 3$. The action (4) is the Abelian-Higgs model [52,53]. It admits a local $U(1)$ symmetry

$$\phi \to \phi\, e^{i\alpha(x)}, \quad \bar{\phi} \to \bar{\phi}\, e^{-i\alpha(x)}, \quad A_a \to A_a + \partial_a\alpha(x)/q. \quad (5)$$

The bulk action (4) is dual to a conformal field theory on the boundary with Hamiltonian $H_{\text{CFT}}$. The local $U(1)$ symmetry of the bulk action corresponds to the global $U(1)$ symmetry of the boundary Hamiltonian $H_{\text{CFT}}$. We identify the global $U(1)$ symmetry with charge conservation.

We consider the field $\phi$ dual to an operator $\mathcal{O}$, and $\bar{\phi}$ dual to $\mathcal{O}^\dagger$. We introduce sources $M$ and $\bar{M}$ for these operators, which corresponds to a deformation of the original Hamiltonian $H_{\text{CFT}}$,

$$H = H_{\text{CFT}} - \int d^{d-1}x\,(M\mathcal{O}^\dagger + \bar{M}\mathcal{O}). \quad (6)$$

If both $\phi$ and $\bar{\phi}$ are related by complex conjugation, we have $M^* = \bar{M}$, and the deformed Hamiltonian (6) is Hermitian. The Gubser-Klebanov-Polyakov-Witten (GKPW) prescription relates the partition functions of the bulk and boundary theory and in the large $N$ and strong coupling limit, it reduces to a saddle point approximation,

$$Z[M,\bar{M}] = \text{Tr}\left[e^{-\beta H_{\text{CFT}} + \int d^dx(M\mathcal{O}^\dagger + \bar{M}\mathcal{O})}\right] \approx e^{-S_{\text{ren}}}\big|_{\phi \to z^{d-\Delta}M, \bar{\phi} \to z^{d-\Delta}\bar{M}}. \quad (7)$$

Here we adopt standard quantization in which $\Delta$ is the scaling dimension of $\mathcal{O}$ given by one of the solutions of $\Delta(\Delta - d) = m^2L^2$, $z$ is the radial coordinate, and $z \to 0$ is the conformal boundary. A nonzero source, either $M$ or $\bar{M}$, breaks the $U(1)$ symmetry (5) and thus, charge conservation is violated on the boundary theory.

To explore non-Hermitian holography, we assume that the GKPW relation (7) still holds even for $M^* \neq \bar{M}$, and hence $\phi^\star \neq \bar{\phi}$, while the holographic dictionary $\phi \leftrightarrow \mathcal{O}$, $\bar{\phi} \leftrightarrow \mathcal{O}^\dagger$ is preserved. Since $H \neq H^\dagger$, there are different ways to define the time evolution. We start with the Euclidean GKPW relation (7), in which the partition function is evaluated by the on-shell action on an Euclidean spacetime. In holography, the time evolution is usually chosen as follows:

$$\langle O(\tau)_E\rangle = \text{Tr}[O(\tau)_E e^{-\beta H}]/\text{Tr}[e^{-\beta H}], \quad O(\tau)_E = e^{\tau H}Oe^{-\tau H}. \quad (8)$$

In order to move to Lorentzian signature, one can perform a Wick rotation $\tau = it$ on the Euclidean spacetime. The observable $O$ measured on the asymptotic boundary of the Lorentzian spacetime at time $t$, has therefore the the expectation value

$$\langle O(t)\rangle = \text{Tr}[O(t)e^{-\beta H}]/\text{Tr}[e^{-\beta H}], \quad O(t) = e^{iHt}Oe^{-iHt}. \tag{9}$$

Hence, the ordinary time evolution considered in holography is the analytical continuation of the Euclidean time path integral in the GKPW relation, which is different from other frameworks of non-unitary evolution in non-Hermitian systems [60, 61]. In Sec. 5, we will comment on the realization of these evolution schemes in holography. Nevertheless, if the theory has time translational symmetry, the one-point functions in (8) and (9) will be time independent, which is consistent with the construction of time-translational solutions in the bulk in this as well as previous works [1, 38].

On the other hand, the conjugation relation between the expectation values $\langle \mathcal{O}\rangle$ and $\langle \mathcal{O}^\dagger\rangle$ does not necessarily hold for $H \neq H^\dagger$, since

$$\langle \mathcal{O}\rangle^* = \text{Tr}[\mathcal{O}^\dagger e^{-\beta H^\dagger}]/\text{Tr}[e^{-\beta H^\dagger}], \quad \langle O^\dagger\rangle = \text{Tr}[\mathcal{O}^\dagger e^{-\beta H}]/\text{Tr}[e^{-\beta H}]. \tag{10}$$

The generalization to $M^* \neq \bar{M}$ can be implemented by the global complexified $U(1)$ transformations

$$\phi \to \phi\, e^{-\theta}, \quad \bar{\phi} \to \bar{\phi}\, e^{\theta}, \quad A_a \to A_a, \tag{11}$$

$$M \to Me^{-\theta}, \quad \bar{M} \to \bar{M}e^{\theta} \tag{12}$$

with $\theta \in \mathbb{C}$. Both the bulk action (4) and also the boundary condition in (7) are invariant under (11) and (12). From the GKPW relation, the partition function is invariant under the transformation (11) and hence, is only a function of the invariant of (11), namely

$$Z[M, \bar{M}] = Z[e^{-\theta}M, e^{\theta}\bar{M}] = Z[N^2], \quad N^2 = M\bar{M}. \tag{13}$$

This means that each value of the invariant $N^2$ labels a class of theories related by the complexified $U(1)$ transformation (12). Denoting the generator of the global $U(1)$ transformation as $Q$, the transformation (11) can be achieved via the similarity transformation

$$H_\theta = e^{\theta Q}He^{-\theta Q} = H_{\text{CFT}} - \int d^dx\, (Me^{-\theta}O^\dagger + \bar{M}e^{\theta}O), \tag{14}$$

where the sources transform in the same way as in (12). So we call it the Dyson map as well. $H_\theta$ will in general be non-Hermitian even though $H$ is Hermitian. Then, the evolution operator $U_\theta = e^{-iH_\theta t}$ could be non-unitary even though the evolution operator $U = e^{-iHt}$ is unitary. Still, the two evolution operators are similar via the Dyson map (14). The similarity transformation preserves the trace and the eigenvalues, so it leaves the partition function (13) invariant. Thus, all the theories with $M\bar{M} \geq 0$ have entirely real eigenvalues, since they are similar to a theory with $M^* = \bar{M}$, whose Hamiltonian is Hermitian.

However, there are more general choices of $M, \bar{M}$ with a real invariant $N^2$, which can be either $N^2 \geq 0$ or $N^2 < 0$. Since $M\bar{M}$ is invariant under the Dyson map (14), the case $N^2 < 0$ can not be mapped to a Hermitian Hamiltonian by the Dyson map (14), which would require $N^2 = M\bar{M} > 0$. Thus, $N^2 = 0$ is the exceptional point. We now show that in all these cases, the holographic theory is $\mathcal{PT}$-symmetric with a proper parity $\mathcal{P}$. See App. A for a fermion model example with $\mathcal{PT}$ symmetry.

Firstly, given a theory with $M, \bar{M} \in \mathbb{R}$, it is $\mathcal{PT}$-symmetric with the following transformation rules of $\mathcal{P}, \mathcal{T}$ and $\mathcal{C}$ [1, 38]

|  | $A$ | $\phi$ | $\bar{\phi}$ | $i$ | $x_1$ | $t$ |
|---|---|---|---|---|---|---|
| $\mathcal{P}$ | $-A$ | $\bar{\phi}$ | $\phi$ | $i$ | $-x_1$ | $t$ |
| $\mathcal{T}$ | $A$ | $\bar{\phi}$ | $\phi$ | $-i$ | $x_1$ | $-t$ |
| $\mathcal{C}$ | $-A$ | $\phi$ | $\bar{\phi}$ | $i$ | $x_1$ | $t$ |

$$(15)$$

without exchanging the sources $M, \bar{M}$. In other words, after the action of $\mathcal{P}$, $\phi$ is sourced by $\bar{M}$ and $\bar{\phi}$ is sourced by $M$; after $\mathcal{T}$, $\phi$ is sourced by $\bar{M}^*$ and $\bar{\phi}$ is sourced by $M^*$. If $M, \bar{M} \in \mathbb{R}$, then both the bulk action and the boundary condition are invariant under this $\mathcal{PT}$ transformation. A theory with $M, \bar{M} \in \mathbb{R}$ defines a 'standard' $\mathcal{PT}$ frame with respect to the complexified $U(1)$ transformation, as it entails a 'standard' definition of the $\mathcal{PT}$ transformation, defined in (15).

Secondly, given a theory with $M, \bar{M} \notin \mathbb{R}$, but $M\bar{M} \in \mathbb{R}$, we can parameterize the sources as $M = M_0 \, e^{-i\theta'}$ and $\bar{M} = \bar{M}_0 \, e^{i\theta'}$ with $M_0, \bar{M}_0, \theta' \in \mathbb{R}$. Then, this theory is exactly the image of the theory with $M_0, \bar{M}_0$ under the complexified $U(1)$ transformation (12) with angle $\theta = i\theta'$. The former Hamiltonian $H'$ is similar to the latter Hamiltonian $H$ via $H' = e^{i\theta' Q} H e^{-i\theta' Q}$. Since $H = \mathcal{PT} H \mathcal{PT}$ and $\mathcal{PT} Q \mathcal{PT} = Q$ [3], $H'$ is invariant under a new $\mathcal{P}'\mathcal{T}$ transformation with $\mathcal{P}' = e^{2\theta' Q} \mathcal{P}$. Starting from the 'standard frame' with $M, \bar{M} \in \mathbb{R}$, in any other complexified $U(1)$ frame, there will be a correspondingly transformed definition of $\mathcal{PT}$ under which the theory is invariant in this frame.

We conclude that each value of the invariant $M\bar{M} = N^2 \in \mathbb{R}$ labels a class of $\mathcal{PT}$-symmetric theories related by the complexified $U(1)$ transformation. Without loss of generality, we can pick a representative in each class with a gauge [1]

$$M = \bar{M} = N. \tag{16}$$

In this gauge, we will show that $\phi = \bar{\phi}$ holds for the solutions in all three phases.

Now we discuss the $\mathcal{PT}$ symmetry of the solution. Similarly, to $M\bar{M}$ labelling a class of Hamiltonians, each profile of $\phi\bar{\phi}$ labels a class of solutions related by complexified $U(1)$ transformation (11) in the bulk. The $\mathcal{PT}$ transformation (15) maps $\phi\bar{\phi} \to \phi^*\bar{\phi}^*$. Thus, given a solution with $\phi\bar{\phi} \in \mathbb{R}$, the $\mathcal{PT}$ symmetry is preserved by the solution; given a solution with $\phi\bar{\phi} \notin \mathbb{R}$, the $\mathcal{PT}$ symmetry is spontaneously broken by the solution.

Last but not least, by setting $\theta = \log(M^*/\bar{M})$ in the similar transformation (14), we get $H_\theta = H^\dagger$. Therefore, the Hamiltonian (6) is pseudo-Hermitian.

## 2.2 Equations and ansatz

We now discuss in detail the equations of motion and the ansatz for the background solution. In our numerical calculation, we set the mass of scalar field to be $m^2 L^2 = -2$ and $L = 1$ to ensure an analytic FG expansion near the conformal boundary. In addition, we choose the coupling coefficient $v = 3/2$. The equations of motion read as follows,

$$R_{ab} + \frac{1}{2} g_{ab} \left( \frac{d(d-1)}{L^2} - m^2 \bar{\phi}\phi - v\bar{\phi}^2\phi^2 \right)$$
$$- D_{(a}\phi D_{b)}^\dagger \bar{\phi} + \frac{1}{2} \left( \frac{1}{4} g_{ab} F_{cd} F^{cd} - F_{ac} F_b{}^c \right) = 0, \tag{17}$$

$$\nabla_a F^{ab} + iq \left( \phi D^{\dagger b} \bar{\phi} - \bar{\phi} D^b \phi \right) = 0, \tag{18}$$

$$DD\phi - m^2\phi - 2v\bar{\phi}\phi^2 = 0, \tag{19}$$

$$D^\dagger D^\dagger \bar{\phi} - m^2\bar{\phi} - 2v\bar{\phi}^2\phi = 0. \tag{20}$$

where we have made use of the on-shell trace of the Einstein equations.

---

[1] In Ref. [1], the authors parameterized the complexified $U(1)$ transformation (12) as $e^\theta = \sqrt{\frac{1+x}{1-x}}$ and the invariant as $N^2 = (1-x^2)\widetilde{M}^2$, where $x$ and $\widetilde{M}$ are real numbers. Thus, changing $x$ with fixed $\widetilde{M}^2 \neq 0$ in their paper is simply changing $N^2$ along the real axis in our paper. Especially, their regions $x^2 < 1$, $x^2 = 1$, and $x^2 > 1$ correspond to our regions $N^2 > 0$, $N^2 = 0$, and $N^2 < 0$, respectively.

In this paper, we investigate static and translationally invariant solutions for the background, and hence the following metric and gauge field ansatz [1]

$$ds^2 = \frac{1}{z^2}\left[-u(z)e^{-\chi(z)}dt^2 + \frac{dz^2}{u(z)} + d\mathbf{x}^2\right], \quad A = A(z)dt. \tag{21}$$

Since we choose $m^2L^2 = -2$, the scaling dimension of the dual scalar operator can have two cases $\Delta = 1$ or $2$, which depends on the choice of quantization. Here we work in standard quantization, and thus identify $\Delta = 2$. In this setup, the source is identified as the leading coefficient in the asymptotic expansion form of the scalar field

$$\phi = Mz + \langle\mathcal{O}\rangle z^2 + \cdots, \quad \bar\phi = \bar M z + \langle\mathcal{O}^\dagger\rangle z^2 + \cdots. \tag{22}$$

If working in alternate quantization with $\Delta = 1$, one should do the exchange $M \leftrightarrow \langle\mathcal{O}\rangle$, $\bar M \leftrightarrow \langle\mathcal{O}^\dagger\rangle$. Since neither $\mathcal{P}$ nor $\mathcal{T}$ exchanges the sources $M, \bar M$, the expansion after the $\mathcal{P}$ transformation is

$$\bar\phi = Mz + \langle\mathcal{O}^\dagger\rangle z^2 + \cdots, \quad \phi = \bar M z + \langle\mathcal{O}\rangle z^2 + \cdots, \tag{23}$$

and after the $\mathcal{T}$ transformation is

$$\bar\phi = M^* z + \langle\mathcal{O}^\dagger\rangle z^2 + \cdots, \quad \phi = \bar M^* z + \langle\mathcal{O}\rangle z^2 + \cdots. \tag{24}$$

Clearly, a nonzero source $M$ or $\bar M$ explicitly breaks the $U(1)$ symmetry (5). For a static solution, we note that the $z$ component of the Maxwell equation (18) requires

$$\phi\partial_z\bar\phi - \bar\phi\partial_z\phi = 0. \tag{25}$$

Thus, the Ward identity

$$\partial_\mu \langle J^\mu\rangle = iq\left(M\langle\mathcal{O}^\dagger\rangle - \bar M\langle\mathcal{O}\rangle\right) = 0 \tag{26}$$

vanishes. $J_\mu$ is the charge current and $\mu$ is the coordinate index on the boundary. The Ward identity (26) can also be obtained by taking the derivative of (13) w.r.t. $\theta$. The vanishing of (26) in spite of the explicit $U(1)$ symmetry breaking is due to the fact that the divergence of the $U(1)$ current is evaluated on a static solution.

Notice that the complexified $U(1)$ invariant combination is $\phi\bar\phi = M\bar M z^2 + \cdots$. As explained in detail in Sec. 2.1, in order to label the equivalence class of partition function $Z[N^2]$, we define $M\bar M = N^2$ with the condition $N = \sqrt{M\bar M} \in \mathbb{C}$. Without loss of generality, we can still rotate the sources to be $M = \bar M = N$ utilizing the complexified $U(1)$ transformation. By this rotation, both the equations of motion and boundary condition are invariant under the exchange of $\phi \leftrightarrow \bar\phi$. With (25), we can further consider the following ansatz for the scalar fields

$$\phi = \bar\phi = \varphi(z) \in \mathbb{C}. \tag{27}$$

where $\varphi(z)^2 = \phi\bar\phi$ is invariant under the complexified $U(1)$ transformation. The ansatz (21)(27) is invariant under the rescaling

$$(z, t, \mathbf{x}) \to (\lambda z, \lambda t, \lambda\mathbf{x}), \quad (u, \chi, A, \varphi) \to (u, \chi, \lambda^{-1}A, \varphi). \tag{28}$$

Substituting the above ansatz into the equations of motion, we obtain four independent equations

$$-\frac{A^2q^2\varphi^2e^\chi z}{u^2} + \chi' - z\varphi'^2 = 0, \tag{29a}$$

$$- \frac{A^2 q^2 \varphi^2 e^\chi z}{2u^2} + \frac{-e^\chi z^4 A'^2 + 4\varphi^2 - 2v\varphi^4 + 12(1-u)}{4uz} - \frac{1}{2}z\varphi'^2 + \frac{u'}{u} = 0, \qquad (29b)$$

$$\frac{\varphi}{u^2}\left(A^2 q^2 e^\chi + \frac{2u}{z^2}\right) - \frac{2v\varphi^3}{uz^2} + \varphi'\left(\frac{u'}{u} - \frac{\chi'}{2} - \frac{2}{z}\right) + \varphi'' = 0, \qquad (29c)$$

$$A'' + \frac{A'\chi'}{2} - \frac{2Aq^2\varphi^2}{uz^2} = 0, \qquad (29d)$$

and one constraint [51]

$$Q_T = \frac{1}{4\pi}e^{\chi/2}\left(AA' - \frac{1}{z^2}(ue^{-\chi})'\right), \qquad (30)$$

whose value on the horizon will give the Hawking temperature $T$. The null energy condition (NEC) is given by

$$T_z^z - T_t^t = \frac{z^2}{u}\left[A^2 q^2 \varphi^2 e^\chi + u^2 (\varphi')^2\right] \geq 0, \qquad (31)$$

which could be violated once $\varphi^2$ becomes negative.

We consider the asymptotic boundary conditions

$$u = 1 + \frac{1}{2}N^2 z^2 + u_3 z^3 + \cdots, \quad \chi = \frac{1}{2}N^2 z^2 + \frac{4}{3}N\langle O\rangle z^3 + \cdots, \qquad (32)$$
$$A = \mu - \rho z + \cdots, \quad \varphi = Nz + \langle O\rangle z^2 + \cdots,$$

with $\mu$ the chemical potential, $\rho$ the charge density, and $u_3$ is related to the energy density, and we have normalized the expectation value $\langle O\rangle$ according to the convention in [52]. So $\langle O\rangle^2 = \langle \mathcal{O}\rangle\langle \mathcal{O}^\dagger\rangle$ is invariant under the complexified $U(1)$ transformation. At finite temperature, we denote the horizon as $z_h$ and impose regularity there, namely,

$$u(z_h) = 0, \quad \chi(z_h) = \chi_h, \quad A(z_h) = 0, \quad \varphi(z_h) = \varphi_h. \qquad (33)$$

The Hawking temperature is defined by

$$T = -\frac{1}{4\pi}e^{-\chi/2}u'|_{z=z_h}, \qquad (34)$$

where $u'$ refers to $\partial_z u(z)$. The entropy density is $s = 4\pi/z_h^2$. From the holographic renormalization given in App. F, which follows from [62–65], the grand canonical potential density $\omega_G$, free energy density $f$, and energy density $\varepsilon$ can be formulated as

$$\omega_G = f - \mu\rho = \varepsilon - Ts - \mu\rho = -\frac{1}{3}(Ts + \mu\rho + 2N\langle O\rangle), \qquad (35)$$

which we find more convenient for numeric calculation. Following the rescaling transformation (28), both the sources and observables change as

$$(N, \langle O\rangle, \mu, T, s, \varepsilon) \rightarrow (\lambda^{-1}N, \lambda^{-2}\langle O\rangle, \lambda^{-1}\mu, \lambda^{-1}T, \lambda^{-2}s, \lambda^{-3}\varepsilon). \qquad (36)$$

Hence, it is convenient to rescale $z_h$ to be unit and parameterize the solutions with dimensionless ratios $(N^2/T^2, \mu/T)$. We present both the neutral case with $\mu = 0$ and the charged case with $\mu \neq 0$ in the following two subsections.

## 2.3 Neutral background

In this subsection, we construct and characterize the three different phases of the holographic model (4) for vanishing chemical potential $\mu = 0$, and finite $\mathcal{PT}$ deformation sourced by $N/T$. We find three different phases, labelled by I, II and III. While the phases I and II have been found already in [1], the finding of phase III is novel.

### 2.3.1 Fixed points and zero temperature solutions

Prior to our analysis at finite temperature, we examine the fixed point and RG structure at zero temperature. In the neutral case with $\mu = 0$, the Maxwell equations (29c) with the boundary conditions $A(0) = A(z_h) = 0$ are solved by $A(z) = 0$ globally.

We first analyze the fixed points structure and find the zero temperature solution. According to [51,66], the model (4) at zero chemical potential has three AdS$_4$ fixed points,

$$\text{UV}: \quad u = 1, \quad \chi = 0, \quad \varphi = 0, \tag{37}$$

$$\text{IR}_\pm: \quad u = 1 + \frac{1}{6v}, \quad \chi = 0, \quad \varphi = \pm\frac{1}{\sqrt{v}}. \tag{38}$$

The UV fixed point is dual to an AdS$_4$ space. Since $\Delta = 2 < 3$, the deformation in (6) is relevant. The IR fixed points are dual to two AdS$_4$ spaces with constant scalar field. The IR$_\pm$ are related by the complexified $U(1)$ transformation (11), which could be reached by the RG flow triggered by a nonzero source $N$.

The interpolation between these fixed points is given by the solutions at zero temperature. We can numerically solve the equations of motion (29) in the $|N/T| \gg 1$ limit or directly work in the domain-wall coordinate [51, 66, 67]. In Fig. 2, the solutions for the scalar field are plotted in the complex $\varphi^2$ plane, i.e. in an invariant way under the complexified $U(1)$ transformation (11). We find one real solution for $N^2 > 0$ and two complex conjugate solutions for $N^2 < 0$, reproducing the result of [1]. Our numeric calculation yields the free energy density, which coincides with the energy density at zero temperature, $f \approx 0.89N^3$ and $(0.89N^3)^*$ in the two branches respectively. The free energy is real when $N^2 > 0$ and imaginary when $N^2 < 0$. The real solution preserves the $\mathcal{PT}$ symmetry and the complex solutions break the $\mathcal{PT}$ symmetry. We will see that they are, respectively, the zero temperature limits of phase I and phase III defined in the next section. The real solution preserves NEC, and the complex solutions make the left-hand side of (31) complex, while the fixed points always satisfy the c-theorem $c_{\text{UV}} \geq c_{\text{IR}}$.

Besides these three solutions, there are families of solutions in the complex plane that can interpolate the fixed points but do not satisfy the boundary condition $N^2 \in \mathbb{R}$. These are related, via the complexified $U(1)$ transformation (12), to the $N^2 \in \mathbb{R}$ case.

### 2.3.2 Phase diagram at finite temperature

We also numerically solve the equations of motion (29) at finite temperature. In virtue of the rescaling symmetry (28), we can fix the horizon radius to $z_h = 1$. This allows to determine the solutions as functions of only the radial coordinate $z$ and the dimensionless ratio $N/T$. We perform the numerical integration of the equations (29) with the boundary conditions (32) and (33) at a specific value of $N$. The temperature $T$ is determined from (34). We investigate the phase structure by varying the values of $N^2$, and plot the expectation value for the scalar operator $\langle O \rangle$, free energy density $f$, and energy density $\varepsilon$ in Fig. 3. We find the following phase structure:

**Phase I** In the region $N^2 \geq 0$, we find one branch of real solutions, which manifestly preserves the $\mathcal{PT}$ symmetry. The expectation value $\langle O \rangle$ is real and negative. The solution in the $N/T \ll 1$ limit coincides with the analytical approximation given in Sec. 2.3.3. The solution in the $N/T \gg 1$ limit asymptotes to the real solution at zero temperature given in Sec. 2.3.1. In addition, NEC is also preserved for all values of $N/T$. The observed increase in the energy density $\varepsilon$ and the free energy density $f$ with increasing $N/T$ is expected, since we introduce a source in the boundary field theory.

**Phase II** In the region $(N/T)_c^2 \leq (N/T)^2 < 0$, there exist two branches of solutions with real metric and imaginary scalar field values [1]. The expectation value $\langle O \rangle$ is purely

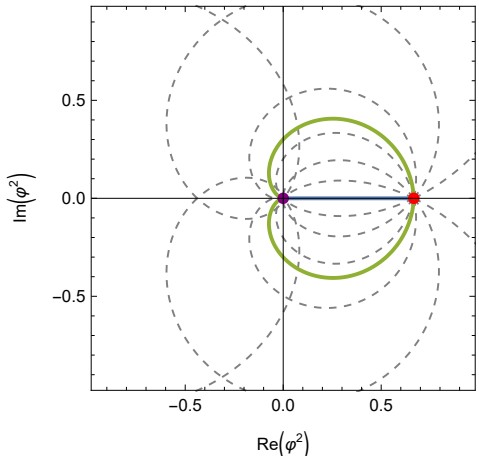

Figure 2: The flows of $\varphi(z)^2$ at zero temperature in the complex plane. They are interpolating between the UV fixed point (purple point) and the two IR fixed points (red point). The flow denoted by the blue curve satisfies the boundary condition $N^2 > 0$. The complex conjugate flows denoted by the two green curves satisfy the boundary condition $N^2 < 0$. The dashed curves denote some of the other flows interpolating the two fixed points but not satisfying the boundary condition $N^2 \in \mathbb{R}$.

imaginary. Even though $\phi\bar{\phi}$ is negative, these solutions still preserve $\mathcal{PT}$ symmetry. For $v = 3/2$, we numerically find a critical ratio $(N/T)_c^2 \approx -3.6$. As in [1], these two branches are both unstable in the sector of scalar $(A_x, \varphi)$ perturbations. In addition, NEC is violated in this case. Interestingly, the branch connected to phase I has a higher free energy density, but a smaller energy density in comparison to the other branch.

**Phase III** In the region of $(N/T)^2 < (N/T)_c^2$, we do not find any solution with real $\varphi^2$ and real metric. However, if we allow the fields to take complex values, we do find a pair of complex conjugate solutions with complex $\varphi^2$ and complex metric, for which $\mathcal{PT}$ symmetry is spontaneously broken. We extract two complex conjugate temperatures $T$, $T^*$ from imposing regularity on the horizon. The solutions in the $|N/T| \gg 1$ limit asymptote to the two complex solutions at zero temperature discussed in Sec. 2.3.1. The expectation value $\langle O \rangle$, free energy $f$, and energy density $\varepsilon$ are complex. Also, the left-hand side of NEC (31) becomes complex. From the quasi-normal mode (QNM) analysis presented in App. (B), the two complex conjugate branches are also unstable.

We now discuss the complex metrics and complex temperatures in phase III in detail. For both solutions in phase III, the metric we calculate numerically has the following asymptotic behavior

$$ds^2 \rightarrow \begin{cases} \left(-dt^2 + dz^2\right)/z^2, & z \to 0 \\ \frac{1}{-u'(z_h)}\left[-(z_h - z)(4\pi T)^2 dt^2 + \frac{dz^2}{z_h - z}\right], & z \to z_h \end{cases}. \tag{39}$$

Here $T \in \mathbb{C}$, $\mathrm{Re}(T) > 0$, and we omit the spatial coordinates. From imposing regularity at the horizon, the coordinate time $t$ acquires a complex period, $t \sim t + i\beta$ with $\beta = 1/T$. If we define a new Euclidean time direction $\tau = i2\pi Tt$ such that it has periodicity $\tau \sim \tau + 2\pi$,

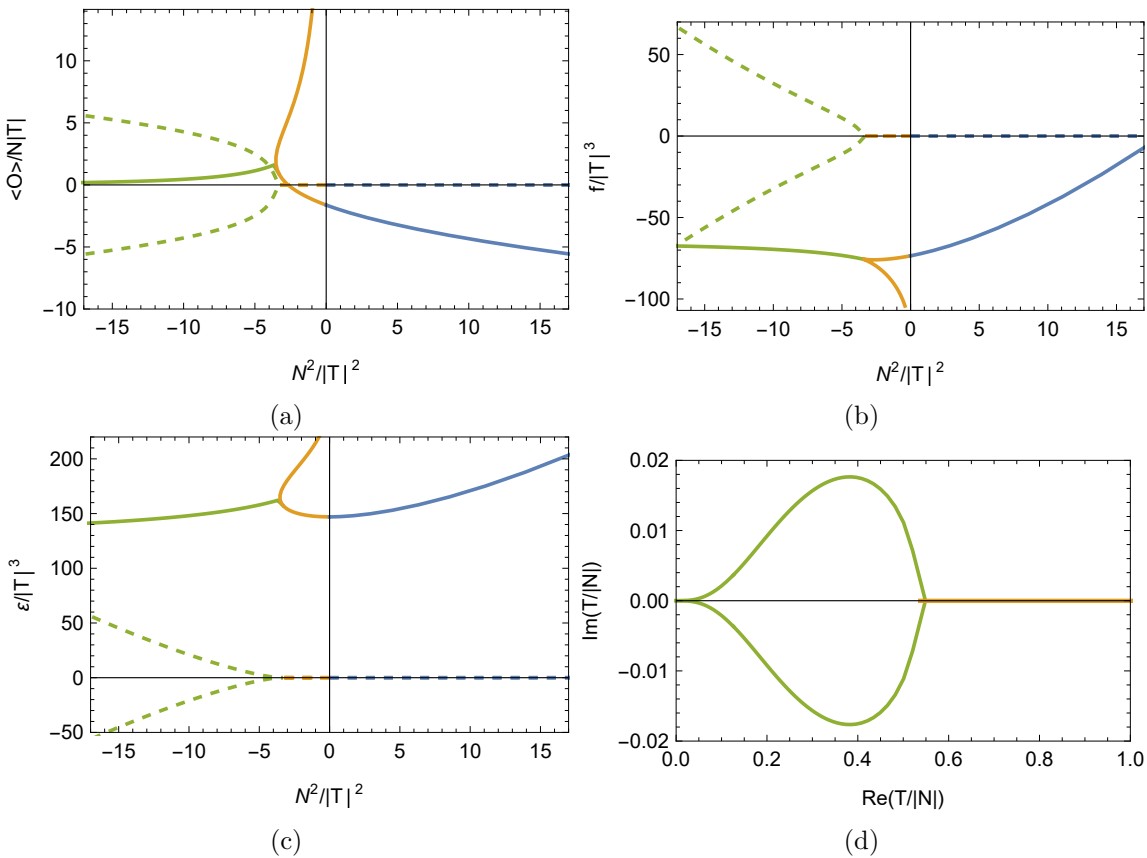

Figure 3: $\langle O \rangle /N\,|T|$ (a), $f/|T|^3$ (b), and $\varepsilon/|T|^3$ (c) as multivalued functions of $N^2/|T|^2$, whose real (imaginary) part is denoted by solid (dashed) curves. The trajectories of $T/|N|$ in its complex plane are shown in panel (d). The results in phases I, II, and III are represented by blue, orange, and green curves respectively.

the metric becomes

$$ds^2 \to \begin{cases} \left[(\beta/2\pi)^2 d\tau^2 + dz^2\right]/z^2, & z \to 0 \\ \frac{1}{-u'(z_h)}\left[4(z_h - z)d\tau^2 + \frac{dz^2}{z_h - z}\right], & z \to z_h \end{cases}, \quad \beta \in \mathbb{C}, \ \mathrm{Re}(\beta) > 0. \tag{40}$$

We see that the line element on a boundary cut-off slice is now complex,

$$\frac{d\tau_{\mathrm{bdy}}^2}{\epsilon^2} = \frac{\beta^2 d\tau^2}{(2\pi\epsilon)^2}, \tag{41}$$

where $\tau_{\mathrm{bdy}}$ is the boundary time and $\epsilon$ is the UV cutoff. Eq. (41) implies a complex inverse temperature $\beta$ in the partition function

$$Z(\beta) = \mathrm{Tr}[e^{-\int_0^\beta d\tau_{\mathrm{bdy}} H}], \quad \beta \in \mathbb{C}. \tag{42}$$

Complex metrics and complex time periods are not exotic in holography. In App. D, we review the "double cone" geometry contributing to the spectral form factor, which also admits complex line element on the boundary, and an interpretation involving a complex temperature.

The two complex conjugate solutions in phase III correspond to partition functions with two complex conjugate inverse temperatures,

$$Z(\beta) = \text{Tr}[e^{-\beta H}], \quad Z(\beta^*) = \text{Tr}[e^{-\beta^* H}], \quad \beta = 1/T \in \mathbb{C}. \tag{43}$$

Their related complex free energies and complex internal energies are shown in Figs. (3b) and (3c). We further checked explicitly that the $|\beta| \to \infty$ limit of the two finite temperature solutions in phase III approach the two zero temperature solutions of [1] discussed in Sec. 2.3.1. Note that the zero temperature limit generally has to be taken along either of the two complex trajectories shown in Fig. 3d corresponding to the two solution branches.

Our holographic interpretation involving complex temperatures in phase III also resolves the following puzzle: The solutions in phase III have complex energies. However, since the $\mathcal{PT}$-symmetric Hamiltonian (6) in holography is pseudo-Hermitian, even in the $\mathcal{PT}$-broken phase, both the partition function $Z(\beta)$, as well as the thermodynamic average of the energy $\langle E \rangle$, should be real for real $\beta$. This apparent contradiction is resolved by concluding that the correct definition of the temperature must be complex in order to avoid a conical singularity at the horizon, and hence both partition functions in (43) are complex conjugate to each other.

The emergence of complex temperatures in the $\mathcal{PT}$-broken phase is also expected in $\mathcal{PT}$-symmetric non-Hermitian systems. This can be seen even in the two-level system analyzed in detail in App. E, where we show that the free energy encounters a branch cut when the temperature is lowered. When the branch cut appears, one should select one branch of the two complex conjugate temperatures, as the $\mathcal{PT}$-broken phase is entered. The zero temperature limit is taken by fixing a nonzero argument of $\beta$ and sending $|\beta| \to \infty$, which reduces both the free energy and the average energy to one of the eigenenergies.

### 2.3.3   The probe limit

We can solve the scalar equation (29d) analytically when the source of the scalar field $N$ in is small compared to the temperature $T$. In this situation, the bulk geometry can be approximated as an $\text{AdS}_4$-Schwarzschild black hole with a scalar field $\varphi$ in the probe limit, i.e. the propagation of this field does not alter the bulk geometry to leading order in $\varphi$. The $\text{AdS}_4$–Schwarzschild solution is given by

$$u(z) = 1 - \frac{z^3}{z_h^3}, \quad \chi(z) = 0, \quad z_h = \frac{3}{4\pi T}. \tag{44}$$

The equation (29d) reduces to

$$\varphi''(z) + \left( \frac{u'(z)}{u(z)} - \frac{2}{z} \right) \varphi'(z) + \frac{2}{z^2 u(z)} \varphi(z) = 0. \tag{45}$$

This equation admits a solution regular at the horizon,

$$\varphi(z) = Nz \left[ {}_2F_1\left(\frac{1}{3}, \frac{1}{3}; \frac{2}{3}; \frac{z^3}{z_h^3}\right) - \frac{2\pi^{3/2} z \, {}_2F_1\left(\frac{2}{3}, \frac{2}{3}; \frac{4}{3}; \frac{z^3}{z_h^3}\right)}{\Gamma\left(\frac{1}{6}\right)^2 \Gamma\left(\frac{7}{6}\right) z_h} \right], \tag{46}$$

with the asymptotic behavior

$$\varphi(z \to 0) = N z - \frac{2\pi^{3/2} N}{z_h \Gamma\left(1/6\right)^2 \Gamma\left(7/6\right)} z^2 + \cdots. \tag{47}$$

From the near boundary expansion, we find

$$\frac{\langle O \rangle}{NT} = -\frac{\sqrt{2}(2\pi)^{5/2}}{3\Gamma\left(1/6\right)^2 \Gamma\left(7/6\right)} \approx -1.63, \tag{48}$$

which matches the numerical result given in Fig. 3a in the limit $|N/T| \ll 1$.

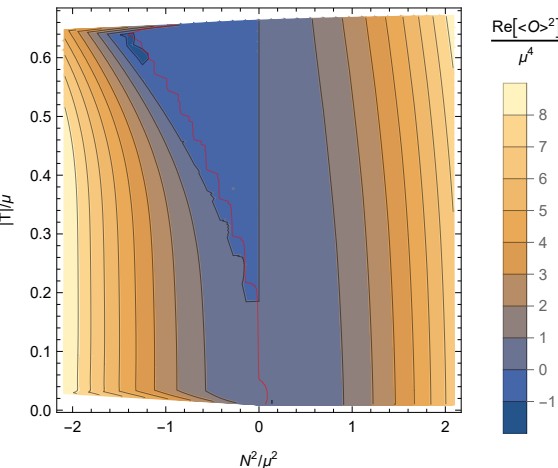

Figure 4: $\text{Re}[\langle O \rangle^2]/\mu^4$ on the plane spanned by $N^2/\mu^2$ and $|T|/\mu$. The red curve denotes the transition between phase II (right) and phase III (left).

### 2.4 Charged backgrounds

In this subsection, we discuss the case of finite chemical potential $\mu$ and derive the charged background solutions numerically. Due to the relation (27) derived from the Maxwell equation, for a given solution at finite $\mu$, the counterpart with $-\mu$ can be obtained just by changing the sign of $A$, while keeping other fields fixed. So, and without loss of generality, we restrict to the $\mu > 0$ case.

From the background solutions, we study the phase structure in the $(N^2/\mu^2, |T|/\mu)$ plane numerically. This is shown in Fig. 4. In particular, we find that the global phase structure above $|T|/\mu \approx 0.02$ is similar to the neutral case in Fig. (1).

Below $|T|/\mu \approx 0.02$, the system has the superconducting instability of the $U(1)$ scalar field of [52,53]. When $N = 0$, this is a second-order phase transition, while for $N^2 > 0$, the transition becomes a cross-over. Since the transition temperature is very small, we do not display it in Fig. 4.

## 3 Conductivity and the sum rule

In this section, we study the linear response of the model (4), and derive the AC conductivity. We consider the Kubo formula of conductivity

$$\sigma^{\mu\nu}(\omega) = \frac{G_R^{\mu\nu}(\omega + i\epsilon)}{i(\omega + i\epsilon)}, \tag{49}$$

$$G_R^{\mu\nu}(t) = -i\theta(t)\frac{\text{Tr}\left[e^{-\beta H}[J^\mu(t), J^\nu]\right]}{\text{Tr}\left[e^{-\beta H}\right]}, \quad J^\mu(t) = e^{iHt}J^\mu e^{-iHt}, \tag{50}$$

where $G_R^{\mu\nu}$ is the retarded Green's function, $\epsilon$ is a positive infinitesimal number, and the time evolution is defined in (9). We will compute the longitudinal conductivity $\sigma(\omega) = \sigma^{xx}(\omega)$ with $G_R(\omega) = G_R^{xx}(\omega)$ and check the validity of the sum rule numerically.

The complexified $U(1)$ transformation (11) does not change the Maxwell field $A$. Thus, the charge current operator $\langle J^\mu \rangle$, and also its correlation function, are both invariant under the Dyson map

$$J^\mu = e^{\theta Q/2}J^\mu e^{-\theta Q/2}, \quad \text{Tr}[e^{iHt}J^\mu e^{-iHt}J^\nu e^{-\beta H}] = \text{Tr}[e^{iH_\theta t}J^\mu e^{-iH_\theta t}J^\nu e^{-\beta H_\theta}]. \tag{51}$$

This invariance also holds for the fermion model presented in App. A. In order to calculate the conductivity from holography, we work in the gauge (16). In addition, notice that the perturbation equation (59) relevant for the conductivity depends on the scalar fields via the complexified $U(1)$ invariant profile $\varphi^2$ only. Therefore, in phase I, we expect the derivation of the AC conducitivity to be equivalent to the Hermitian case. The invariance of Dyson map in holography also supports our definition of evolution and Green function (50).

However, since the theory with $N^2 < 0$ is not similar to a Hermitian theory via Dyson map (14), the conductivity relation

$$\sigma(\omega) = \sigma(-\omega)^* \tag{52}$$

may not necessarily hold in the $N^2 < 0$ region. Remarkably, we will show that (52) is a necessary condition in order for the sum rule to hold.

We firstly check the sum rule (3) by analyzing the properties of the retarded Green's function of the charged current following [55]. As required by causality and the asymptotic behavior of the retarded Green's function for the sum rule to hold, the following two conditions must be met:

1. $G_R(\omega)$ is analytical on the upper half plane and on the real axis;

2. $\lim_{|\omega| \to \infty} G_R(\omega) = i\omega$.

The first condition can be checked from the QNM spectrum in the holographic model, which will be done at the end of this section. The second condition is related to the asymptotic behavior of $G_R(\omega)$ in the high frequency limit, which for this model becomes the current Green's function of the AdS-Schwarzschild black hole without any scalar fields. By applying Cauchy's theorem, we have

$$G_R(\omega + i\epsilon) - i\omega = \int_{-\infty}^{\infty} \frac{dz}{2\pi i} \frac{G_R(z) - iz}{z - \omega - i\epsilon}, \quad 0 = \int_{-\infty}^{\infty} \frac{dz}{2\pi i} \frac{G_R(z) - iz}{z - \omega + i\epsilon}. \tag{53}$$

Adding up the first integral and the complex conjugate of the second integral leads to the spectral representation

$$G_R(\omega) - i\omega = \lim_{\epsilon \to \epsilon} \int_{-\infty}^{\infty} \frac{dz}{\pi} \frac{\text{Im}[G_R(z)] - z}{z - \omega - i\epsilon}. \tag{54}$$

Setting $\omega = 0$ and using the Cauchy principal value integral, we obtain the sum rule (3).

In order to check the sum rule numerically, we also need to introduce the integrated spectral weight $S_\sigma(\Omega)$,

$$S_\sigma(\Omega) = \int_{-\Omega}^{\Omega} (\text{Re}[\sigma(\omega)] - 1) d\omega. \tag{55}$$

In particular, notice that $S_\sigma(\infty) = 0$ is exactly the sum rule (3). In the pure AdS-Schwarzschild case, it vanishes exactly regardless of $\omega$. A non-zero value of (3) would signal a severe breakdown of either causality, unitarity, or charge conservation, due to the $\mathcal{P}$- and $\mathcal{T}$-invariance violation in the model (4).

## 3.1   AC conductivity

As the time evolution (9)(49) realized in AdS/CFT is of the same form as in the Hermitian case, we follow the standard procedure to derive the conductivity within the AdS/CFT correspondence [68] from linear gauge field perturbations. We consider the fluctuations of the gauge field and metric component

$$\delta A_x = a(z)e^{-i\omega t}, \quad \delta g_{tx} = h_{tx}(z)e^{-i\omega t} \tag{56}$$

in the equations (17)-(20). The only non-trivial equations come from the $tx$ component of the Einstein equations and the $x$ component of the Maxwell equations, which read

$$az^2 A' + h'_{tx} = 0, \tag{57}$$

$$4u^2 z^2 a'' + uz a' \left( e^\chi z^4 (A')^2 + 12u - 4\varphi^2 + 2v\varphi^4 - 12 \right)$$
$$+ a \left( 4e^\chi \omega^2 z^2 - 8q^2 u\varphi^2 \right) + 4u e^\chi z^2 A' h'_{tx} = 0. \tag{58}$$

The mix between $a(z)$ and $h_{tx}(z)$ indicates a coupling of charge current and momentum. Solving (57) for $h'_{tx}(z)$ and substituting it in (58), we get a single equation for $a(z)$

$$4u^2 z^2 a'' + uz a' \left( e^\chi z^4 (A')^2 + 12u - 4\varphi^2 + 2v\varphi^4 - 12 \right)$$
$$- 4a \left( e^\chi z^2 \left( uz^2 (A')^2 - \omega^2 \right) + 2q^2 u\varphi^2 \right) = 0 \,. \tag{59}$$

Imposing the ingoing boundary condition at the horizon

$$a(z) = (z_h - z)^{-i\frac{\omega}{4\pi T}} b(z) \,, \tag{60}$$

and requiring the regularity of $b(z)$ near $z_h$, we get the retarded Green's function $G_R^{xx}(\omega)$. From the Kubo formula (49), we get

$$\sigma(\omega) = \frac{a'(0)}{i(\omega + i\epsilon) a(0)} \,. \tag{61}$$

The infinitesimal shift $\epsilon$ accounts for the delta peak at $\omega = 0$ in the real part of the conductivity, which is essential for examining the validity of the sum rule (3). In the high frequency limit, namely $\omega \gg |T|, \mu, |N|$, the conductivity approaches $\sigma \to 1$, as expected for the AdS-Schwarzschild geometry that governs the UV, while the low frequency behavior depends on the IR geometry. The low-frequency behavior is governed by the Kramers-Kronig (KK) relation, with two contributions denoted by $\rho_n$ and $\rho_s$ in such a way that

$$\sigma(\omega) = (\rho_s q^2 + \rho_n) \left( \pi\delta(\omega) + \frac{i}{\omega} \right) + \text{regular terms}, \tag{62}$$

where $\rho_s$ is the superfluid density and $\rho_n$ is the normal charge density. Next, we present the numerical results of the AC conductivity on both the neutral and charged backgrounds and further analyze the asymptotic behaviors at high and low frequencies.

## 3.2 Numerical results

We first calculate the conductivity on the neutral background, i.e. $\mu = 0$. The conductivity $\sigma$ shown in the left panels of Figs. 6, 7 and 8 correspond to phase I, II and III respectively. The right panels of these figures show the convergence of the integral (55), which is used to check the validity of the sum rule.

In the high frequency limit, there exists a common asymptotic behavior for all phases, $\sigma(\omega) \to 1$, or $G_R(\omega) \to i\omega$ for $\omega \to \infty$. This is exactly the conductivity for the asymptotic AdS-Schwarzschild background in the UV regime. In the right panels of Figs. 6, 7 and 8, we also observe that the integral $S_\sigma(\Omega)$ in (55) always exhibits a power-law decay at $\Omega/|N| \gg 1$, which confirms that the sum rule always holds. As an independent check, we also show that the sum rule holds from the quasi-normal modes spectrum at the end of this section.

In the low frequency limit, the conductivity agrees with our expectation (62) with $\rho_n = 0$ due to the neutrality of the background. We extract $\rho_s$ from fitting the numerical conductivity and show it as a function of the square of the condensate $\langle O \rangle^2$ in Fig. 5. In particular, we find $\rho_s \sim \langle O \rangle^2 / T^3$ for small $\langle O \rangle^2$. Therefore, the low frequency behavior of conductivity varies with the particular phase considered, with the following properties:

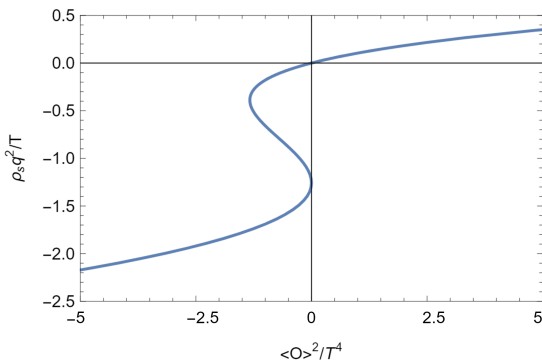

Figure 5: $\rho_s q^2/T$ as a function of $\langle O \rangle^2/T^4$ in phases I ($\rho_s > 0$) and II ($\rho_s < 0$) with $\mu = 0$. Phase III, where $\rho_s q^2/T$ becomes complex and is not shown in this plot, extends from the phase II at $\rho_s q^2/T \approx -1.3$.

**Phase I** In this phase, we find that $\text{Im}\,\sigma$ can be approximated as $\rho_s q^2/\omega$ with $\rho_s > 0$ at low frequencies. Due to the form (62), the conductivity satisfies the relation (52), as required by Hermiticity. In addition, the regular part of $\sigma(\omega)$ fulfills $\text{Re}\,\sigma < 1$. This is required by the existence of a positive weight of the $\delta(\omega)$ function due to the sum rule.

**Phase II** In this phase, $\text{Im}\,\sigma$ can still be approximated as $\rho_s q^2/\omega$ at small $\omega$, except that now $\rho_s < 0$. Since $\rho_s \in \mathbb{R}$, the conductivity still satisfies the relation (52). Furthermore, we observe that the analytical part of $\sigma(\omega)$ fulfills $\text{Re}\,\sigma > 1$, which is also required by the existence of a negative weight of the $\delta(\omega)$ function in the sum rule.

**Phase III** In this phase, we find that $\sigma \approx i\rho_s q^2/\omega$ at small but nonzero $\omega$ with $\rho_s \in \mathbb{C}$. Our numerical analysis finds that $\rho_s \in \mathbb{C}$, and hence both $\text{Re}\,\sigma$ and $\text{Im}\,\sigma$ have $1/\omega$ behavior. Consequently, the conductivity does not satisfy the relation (52). Instead, the conductivity on the one branch of the background is mapped to the one on its conjugated branch under the change $\omega \to -\omega$ by (52). This requires to include $\delta(\omega)$ in both the real and imaginary parts as well.

The $\mathcal{PT}$ symmetry is preserved in phases I and II, but broken in phase III. This breaking leads to the emergence of the complex weight $\rho_s$ and thus, to the breaking of the symmetry (52) of the conductivity under the reflection $\omega \to -\omega$. Still, the sum rule always holds. The analytical study of conductivity in the probe limit $N \ll T$ in App. C also confirms the sum rule and shows that the superfluity density is proportional to the scalar field source, $\rho_s \propto N^2$.

We also investigated the conductivity for finite chemical potential $\mu$. By inspecting Fig. 9, we can observe, as we dial $N/\mu$, a competition between the superfluid density $\rho_s$ and normal charge density $\rho_n$ in the low frequency limit (62) in phase II. The weight $\rho_s q^2 + \rho_n$ can be either negative or positive. Nevertheless, the sum rule always holds, as illustrated in the right plot of Fig. 9. The combination of $\rho_s q^2 + \rho_n$ also appears in phases I and III numerically.

## 3.3 Sum rule from quasi normal mode

We already observe the power law decay of the integral $S_\sigma(\Omega)$ numerically in last subsection. In this subsection, we doubly check the first condition for the sum rule, stability, namely $G_R(\omega)$ being analytic both on the upper half plane and the real axis. The poles of

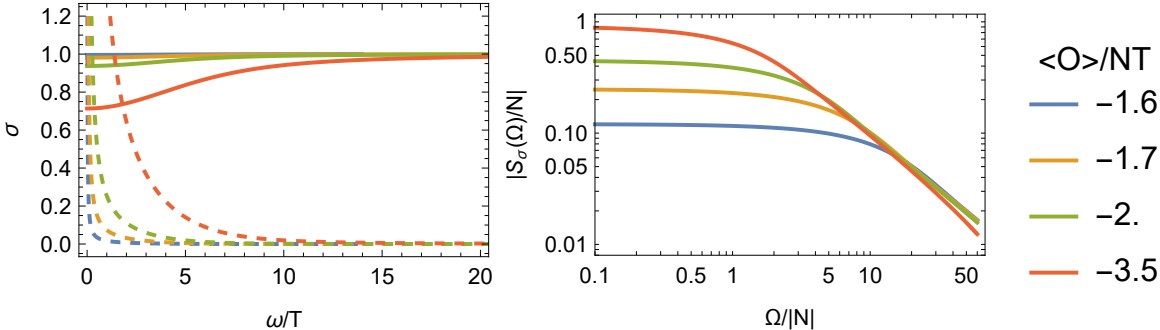

Figure 6: Left: The conductivity in phase I as a function of $\omega/T$. The real (imaginary) part is denoted by solid (dashed) lines. Right: The integral spectral weights (55) as functions of the frequency bound.

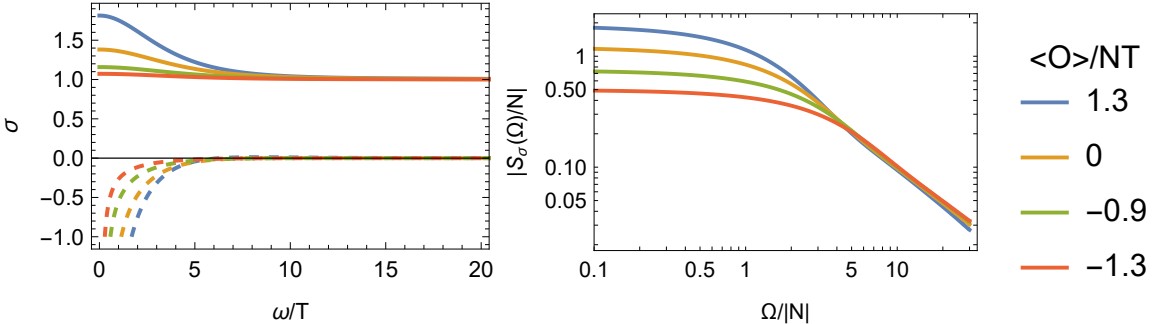

Figure 7: The conductivity and its integral in phase II.

$G_R(\omega)$ are given by the QNMs of the $(A_x, g_{tx})$ components [56]. For our numerical analysis, we arrange the differential equation (59), the boundary conditions (60) and $a(0) = 0$ together as a linear differential operator $\mathcal{D}$ of the form

$$\mathcal{D}[a(z)] = 0. \tag{63}$$

If the determinant of (63) vanishes for a set of frequencies, we found a QNM. One can write the operator $\mathcal{D}$ as a matrix and calculate the $1/\det[\mathcal{D}]$. In Fig. 10, we plot this quantity as a function of the complex frequency, and infer that there is no pole on the upper half plane and the real axis. This shows that the retarded Green's function $G_R(\omega)$ is analytic both on the upper half plane and the real axis. Therefore, the first condition for the sum rule mentioned in Sec. 3 is satisfied. By combining it with the asymptotic condition $G_R(\omega) = i\omega$ for $|\omega| \to \infty$ due to the UV fixed point, we independently confirm that the sum rule holds in all the phases.

## 4 Field theory: $U(1)$ rotor model

To better understand phase III, in which the complex superfluity density emerges in the $\mathcal{PT}$-symmetric holographic model, in this section, we construct an effective model in field theory which reproduces the phase transitions and the zero frequency spectral weight of conductivity found in the holographic model. We start from a $U(1)$ rotor model with a charged scalar $\phi$ coupled to a gauge field $A_\mu$. Then, we allow the scalar field $\phi$ to be independent from its complex conjugate $\bar{\phi}$, such that the original $U(1)$ symmetry is

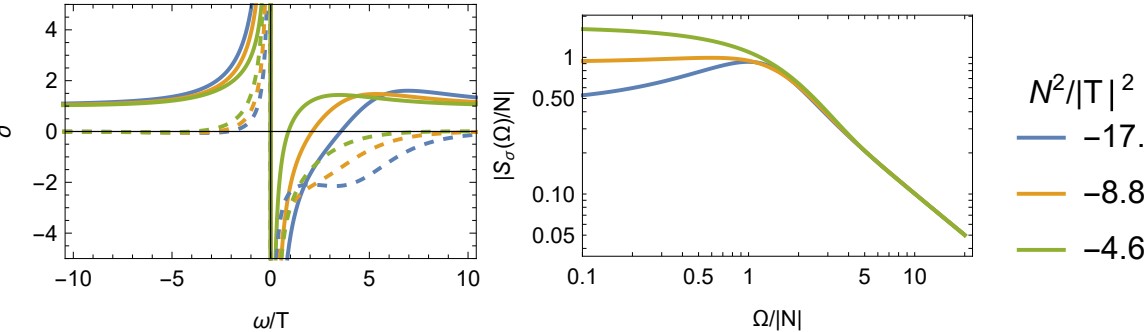

Figure 8: The conductivity and its integral in phase III.

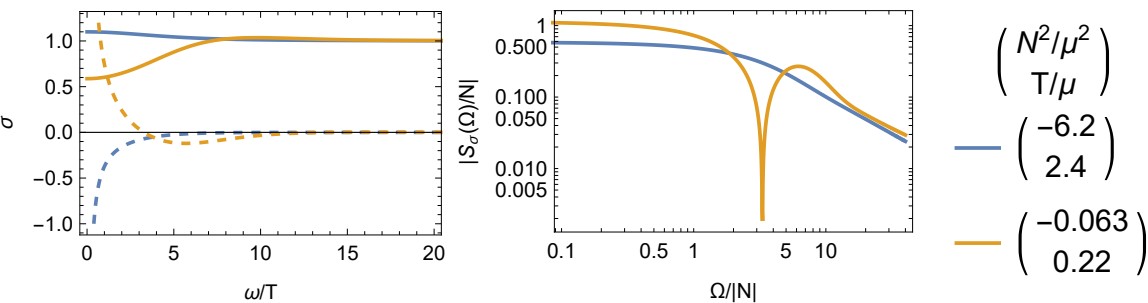

Figure 9: The conductivity and its integral in phase II at finite charge. The spike in the yellow curve in the right panel is due to the integral (55) going through a zero at some finite cutoff frequency $\Omega$, which in the log-log plot leads to an apparent divergence.

complexified. We refer to this model as the complexified $U(1)$ rotor model. After this, we break the complexified $U(1)$ symmetry and also Hermiticity by introducing the $\mathcal{PT}$-symmetric deformations. Its Euclidean action takes the form

$$S_\phi = \int d^d x \left( D_\mu \phi D^{\dagger\mu}\bar\phi + V(\phi, \bar\phi) + \bar M \phi + M\bar\phi \right), \quad V(\phi, \bar\phi) = r\phi\bar\phi + \frac{1}{2}u\phi^2\bar\phi^2, \quad (64)$$

where $D_\mu = \partial_\mu - iqA_\mu$, $\phi$ and $\bar\phi$ are two independent fields, $r$ is the mass square, $u$ is coupling constant, and $M, \bar M$ are two independent sources, respectively. So this model is an analogue of the holographic model (4). Following (15), the application of $\mathcal{T}$ transforms $\phi \leftrightarrow \bar\phi$, $i \leftrightarrow -i$, and the application of $\mathcal{P}$ transforms $\phi \leftrightarrow \bar\phi$. Letting $A_\mu = 0$ and considering a static and translational invariant solution to the scalar fields, the saddle point $\phi_s, \bar\phi_s$ of the potential satisfies the following equations

$$r\phi_s + u\phi_s^2\bar\phi_s + M = 0, \quad r\bar\phi_s + u\bar\phi_s^2\phi_s + \bar M = 0. \quad (65)$$

Similar to the holographic model, we can consider the complexified $U(1)$ transformation,

$$M \to Me^{-\theta}, \quad \bar M \to \bar Me^{\theta}, \quad \theta \in \mathbb{C}. \quad (66)$$

Then, the saddle point solution follows the transformation rules

$$\phi_s \to \phi_s e^{-\theta}, \quad \bar\phi_s \to \bar\phi_s e^{\theta}, \quad (67)$$

such that the on-shell action is invariant. This transformation preserves $M\bar M$ and $\rho_s = 2\phi_s\bar\phi_s$, which enables us to choose $M = \bar M = N \in \mathbb{C}$ without loss of generality. Once

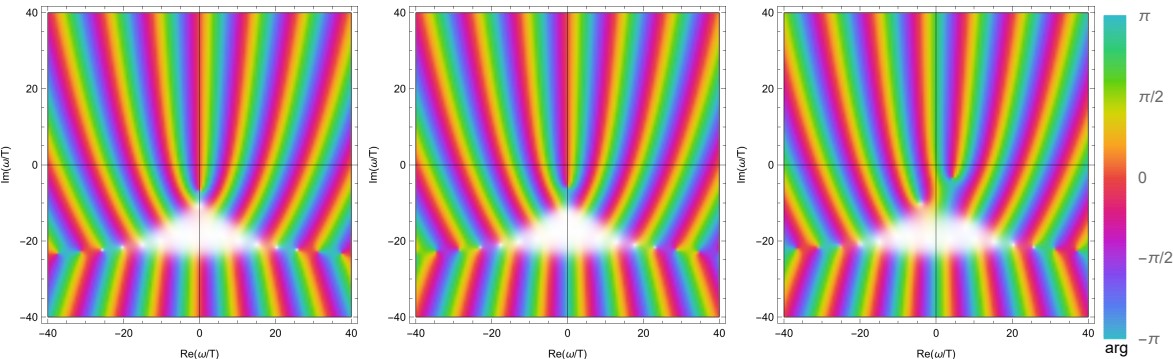

Figure 10: $1/\det[\mathcal{D}]$ in the complex $\omega/T$ plane in phase I, II, and III (from left to right), where the color denotes the argument and the shading denotes the absolute value of $1/\det[\mathcal{D}]$. The white points denote the poles. The three plots correspond to $(N^2/\mu^2, T/\mu) = (6.2, 2.4), (-6.2, 2.4), (-100, 2.4 + 0.1i)$, respectively.

$N \neq 0$, the equations give $\phi_s = \bar{\phi}_s = \varphi_s$. Finally, the on-shell action can be related to a $\mathcal{PT}$-symmetric action via the complexified $U(1)$ transformation if $M\bar{M} = N^2 \in \mathbb{R}$.

We restrict to the $u > 0$ case, in order for the potential to be bounded from below for real $\phi, \bar{\phi}$. When $N = 0$, the action admits the global $U(1)$ symmetry

$$\phi \to \phi\, e^{-i\alpha}, \quad \bar{\phi} \to \bar{\phi}\, e^{i\alpha}. \tag{68}$$

If we set $r < 0$, the potential $V(\phi, \bar{\phi})$ looks like a Mexican hat with three extrema on the real axis. Then the $U(1)$ symmetry will be broken spontaneously by the nonzero solution. However, in holographic approach (4), we consider an explicit breaking of the $U(1)$ symmetry under the transformation (5). Thus, in order to establish an analogy to the holographic model, we consider $r > 0$. This entails that there is only one saddle point of the potential $V(\phi, \bar{\phi})$ on the real axis. To break the $U(1)$ symmetry explicitly and trigger nonzero solution, we turn on the source $N$.

Around the resulting saddle point, we consider phase fluctuations $\phi = \varphi_s e^{-i\alpha}$ and $\bar{\phi} = \varphi_s e^{i\alpha}$. The action in the small $\alpha$ expansion is then given by

$$S_\alpha \approx \frac{1}{2}\rho_s \int d^d x \left[ (\partial_\mu \alpha - qA_\mu)(\partial^\mu \alpha - qA^\mu) - m^2 \alpha^2 \right], \tag{69}$$

where $\rho_s = 2\varphi_s^2$ plays the role of the superfluid density and the mass square of pseudo-Goldstone is

$$m^2 = N/\varphi_s. \tag{70}$$

To study the dynamic stability at $A = 0$, we can consider a general fluctuation around the saddle point $\phi = \varphi_s + \delta\phi(\omega, \vec{k})$, $\bar{\phi} = \varphi_s + \delta\bar{\phi}(\omega, \vec{k})$ and extract their effective mass square matrix $\mathcal{M}^2$. The linear perturbation equation around a saddle point $\varphi_s$ in frequency-momentum domain is

$$(-\omega^2 + \vec{k}^2 + \mathcal{M}^2)\delta\vec{\phi} = 0, \tag{71}$$

where $\delta\vec{\phi} = (\delta\phi, \delta\bar{\phi})^T$. Around a saddle point $\varphi_s$, the negative or complex eigenvalues of $\mathcal{M}^2$ correspond to the dynamically unstable modes.

We always find three saddle points in the complex plane because of the quartic potential. Their movement depending on the value of $N^2$ is shown in Fig. 11. One of the saddle points remains a spectator, which does not merge with the others throughout the domain of parameters we study. We find three different phases as follows:

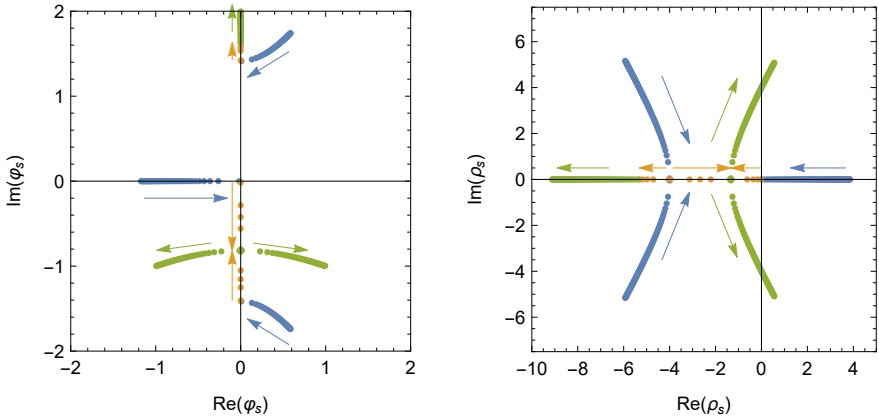

Figure 11: The movement of the saddle point solution $\varphi_s$ (left) and superfluid condensate $\rho_s$ (right) in the complex plane when $N^2$ decreases from 30 to $-30$ and $r = 2$, $u = 1$. The blue dots, orange dots, and green dots denote phase I, phase II, and phase III, respectively. The transition between phase I and phase II happens at $N^2 = 0$ and the transition between phase II and phase III happens at $N_c^2 = -1.18$.

**Phase I** When $N^2 \geq 0$, there is one real solution $\varphi_s$ with $\rho_s > 0$, and two complex conjugate values of $\varphi_s$ with $\rho_s \in \mathbb{C}$. Only the real solution is stable.

**Phase II** When $N_c^2 < N^2 < 0$, there are three imaginary values of $\varphi_s$ with $\rho_s < 0$. Only the solution connected to the real solution in phase I is stable.

**Phase III** When $N^2 < N_c^2$, there is one imaginary $\varphi_s$ with $\rho_s < 0$ and two complex conjugate values of $\varphi_s$ with $\rho_s \in \mathbb{C}$. All solutions are unstable.

The transition between phase II and phase III is due to the collision of two imaginary saddle points, which is analogous to the case in the holographic model.

In all phases, the $U(1)$ current is

$$J^\mu = \frac{\delta S_\alpha}{\delta A_\mu} = -\rho_s q(\partial^\mu \alpha - q A^\mu). \tag{72}$$

The Kubo formula in Euclidean momentum space is

$$K^{\mu\nu}(k) = \frac{\delta \langle J^\mu(k) \rangle}{\delta A^\nu(k)} = -\rho_s q^2 \left( \delta^{\mu\nu} - \frac{k^\mu k^\nu}{k^2 + m^2} \right). \tag{73}$$

After Wick rotating to real time, the superfluid conductivity along the $x$ direction is

$$\sigma_s(\omega) = \frac{K^{xx}(i(\omega + i\epsilon), k = 0)}{i(\omega + i\epsilon)} = \rho_s q^2 \left( \pi \delta(\omega) + \frac{i}{\omega} \right). \tag{74}$$

The integral of the spectral weight is

$$\int_{-\infty}^{\infty} \mathrm{Re}\sigma_s(\omega) d\omega = \pi \rho_s q^2, \tag{75}$$

which depends on the sources $M$ and $\bar{M}$ via $\rho_s$ only. Notice that from (75), no actual violation of the sum rule can be deduced, as we are considering the phase fluctuation at quadratic order in (69) and ignoring the existence of loop corrections coming from both phase and absolute value (Higgs mode) fluctuations, which is the reason for the absence of the normal (not superconducting) contribution to the conductivity. In summary, the rotor model (64) reproduces the behavior of the condensate in all three phases found in the holographic model (4).

# 5 Conclusions

In this work, we studied the solution structure and AC conductivity for the non-Hermitian holographic model of [1], both at zero and at finite density. The solution space we examined consisted of the two sources of the $U(1)$ complex scalar operator $M$ and $\bar{M}$, the temperature $T$, and chemical potential $\mu$. The $\mathcal{PT}$-symmetric non-Hermitian deformation consists of effectively decoupling $M$ and $\bar{M}$, where in the usual Hermitian setup, $\bar{M}$ is the complex conjugate of $M$. The $\mathcal{PT}$ deformation switches on sources for the scalar operators $\langle\mathcal{O}\rangle$ and $\langle\mathcal{O}^\dagger\rangle$, which in the Hermitian case also are complex conjugate to each other. Both operators receive expectation values, i.e. condensates $\langle\mathcal{O}\rangle$ and $\langle\mathcal{O}^\dagger\rangle$. Note that the sources $M$ and $\bar{M}$ break the $U(1)$ symmetry explicitly, and not spontaneously as in the holographic superconductor of [40].

Within the explored solution space, there are two $\mathcal{PT}$-symmetric phases at finite temperature with real (phase I) and imaginary (phase II) condensates, respectively, which have already been constructed in [1]. The finite temperature solutions in phase II approach the exceptional point in a zero temperature limit in which the ratio $N/T$ is held fixed. We furthermore calculated the free energy in phase II (c.f. Fig. 3b), and in this way identified the thermodynamically dominant solution branch. Since the exceptional point in the holographic model [1] is exactly AdS$_4$, the phase diagram Fig. 1 suggests that phase II is actually part of a quantum critical region at finite temperature, which emanates from the quantum critical exceptional point at zero temperature.

We also presented and discussed in detail the emergence of a $\mathcal{PT}$-broken finite temperature phase with complex condensates (phase III). We checked that our finite temperature solutions approach, in the zero temperature limit, the complex conjugate pair of extremal solutions constructed in [1]. A peculiar feature of our finite temperature solutions in phase III is that the metric sourced by the scalar field also becomes complex, while it remained real in phases I and II. Requiring the absence of a conical singularity at the horizon implies that the temperature acquires an imaginary part in phase III. While a complex temperature seems puzzling at first, as discussed in App. D, a similar phenomenon appears in the double cone geometry of JT gravity. In addition, we quantified the transition point from phase II to phase III, which occurs at a critical value of $(N/T)_c \approx -3.6$. We note that in general, this value is a function of the conformal dimension $\Delta = 2$, the charge of the scalar field $q = 1$, and the coefficient of the quartic term in the action $v = 3/2$. For future work, it will be interesting to explore the parameter space $(\Delta, q, v)$ thoroughly in order to further analyze the phase structure of $\mathcal{PT}$ deformed holographic Einstein-Maxwell-scalar theory [51, 69]. In addition, following [70] and analysing quantum critical points from the top-down perspective should be insightful as well.

Furthermore, we calculated the AC conductivity for each of the three phases and observed a shift of spectral weight to a delta function at zero frequency as a function of the $\mathcal{PT}$ breaking parameter $N$. Zero frequency spectral weight can be induced by the condensation of charged operators as well as by normal charge densities, with the latter being absent at zero chemical potential. We found that the zero frequency spectral weight induced by the condensate is positive in phase I, negative in phase II, and complex in phase III, leading to a delta function and a $1/\omega$ pole in both $\mathrm{Re}\,\sigma$ and $\mathrm{Im}\,\sigma$. Still, in all three cases, a mode analysis in the $(A_x, g_{tx})$ channel related to the longitudinal charge transport shows that the Kramers-Kronig relations hold, as all quasinormal modes are in the lower half frequency plane. Moreover, the quantum critical conductivity, which is the DC limit of the real analytic part of the AC conductivity, is suppressed in phase I and enhanced in phase II, as compared to the value $\sigma_Q = 1$ (in units of $e^2/h$) for the AdS-Schwarzschild solution. We also extracted the quantum critical conductivity in phase III, finding a complex value for $\sigma_Q$ due to the breakdown of the relation (52) In contrast, if (52) holds, the imaginary part of $\sigma_Q$ always vanishes. We also observe that the FGT

sum rule always holds from two perspectives: 1) as a direct integral of the real part of the AC conductivity, and 2) from the analysis of the quasi-normal mode spectrum. The sum rule holds both in the $\mathcal{PT}$-symmetric and broken phases, and is in essence responsible for the suppression or enhancement of the quantum critical conductivity due to the spectral weight in the $\delta(\omega)$ pole.

We also constructed and analyzed a complexified $U(1)$ rotor model with an analogous $\mathcal{PT}$-symmetric deformation from effective field theory principles. By tuning the sources of the scalar operators, we find real, imaginary and even complex solutions. The nonzero sources break the $U(1)$ symmetry explicitly, and pseudo-Goldstone bosons with a finite mass appear. We studied the AC conductivity in all phases of the rotor model. We found an analogous shift of spectral weight to the delta functions at zero frequency as in the holographic model. The spectral weight is given by the condensates. The rotor model also admits some differences to the holographic model: First, from (75), we find that the apparent violation of sum rule in all phases, as we only consider the phase fluctuation around the saddle point in the rotor model, instead of solving the whole model, including the absolute value of the scalar field. Second, as evident from the left panel of Fig. 11, in all three phases, the rotor model has one or more additional complex solution compared to the holographic model Fig. 3. However, this does not necessarily mean that the holographic model has another solution that we possibly missed in our numerical analysis. As all the properties of the solutions of the holographic model are reproduced by at most two solutions of the rotor model, it looks as if the additional solution in the rotor model is just a spectator which does not play any role in the universality argument.

For future work, it will be interesting to investigate $\mathcal{PT}$-symmetric non-Hermitian versions of superconductivity in holography [71], and compare to recent field theory studies [72–78], as well as to holographic hydrodynamics [79]. Another interesting route is to further investigate $\mathcal{PT}$-symmetric deformations of SYK-type models, following [80–85]. Finally, it will be interesting to calculate the quantum critical conductivity from kinetic theory in a $\mathcal{PT}$-symmetric Dirac metal along the lines of [86], in order to understand how sensitive is the suppression/enhancement feature we found in the holographic model to the coupling strength.

Finally, in this paper, we use the standard GKPW relation (7) in Euclidean time and analytically continue it to Lorentzian time (9). This defines a particular holographic time evolution in the presence of the $\mathcal{PT}$ deformation. Alternatively, some studies [60, 61, 87] consider a Hermitian density matrix $\rho$ such as the Gibbs state $\rho \propto e^{-\beta H_{\mathrm{CFT}}}$, and define real time evolution as $e^{-iH^\dagger t}\rho e^{iHt}$ with the non-Hermitian Hamiltonian $H$. Correspondingly, the Heisenberg picture of an operator $O$ becomes $O(t) = e^{iH^\dagger t}Oe^{-iHt}$, and the expectation value will in general be time-dependent,

$$\langle O(t) \rangle = \mathrm{Tr}[e^{iH^\dagger t}Oe^{-iHt}\rho] \,. \tag{76}$$

Obviously, this is different from the holographic construction considered here and in [1], where the background solutions are all static, i.e. time-independent. Also, the evolution in (76) will not be invariant under the complexified $U(1)$ transformation, as $e^{\theta Q}H^\dagger e^{-\theta Q} \neq (e^{\theta Q}He^{-\theta Q})^\dagger$ for a general $\theta \in \mathbb{C}$. To compute the expectation value (76) with $\rho \propto e^{-\beta H_{\mathrm{CFT}}}$ and the non-Hermitian Hamiltonian $H$ in (6) in holography, we could first prepare an Euclidean black hole at temperature $1/\beta$ without sources, next apply a quench on one side with $H_{\mathrm{CFT}} + \int d^{d-1}x \,(M\mathcal{O}^\dagger + \bar{M}\mathcal{O})$ for time $t$ and a quench on the conjugate side with $H_{\mathrm{CFT}} + \int d^{d-1}x \,(\bar{M}^*\mathcal{O}^\dagger + M^*\mathcal{O})$ for time $t$, then measure the observable $O$, and finally glue the two sides together following [88, 89]. We leave it for future work [90].

# 6    Acknowledgements

We thank Daniel Areán, Matteo Baggioli, Sebastian Grieninger, Ren Jie, Viktoriia Kornich, Karl Landsteiner, Ronny Thomale and Björn Trauzettel for useful discussions. The work of Z. Y. X., D. R. F., Z. C. and R. M. was funded by the Deutsche Forschungsgemeinschaft (DFG, German Research Foundation) through Project-ID 258499086—SFB 1170 "ToCoTronics" and through the Würzburg-Dresden Cluster of Excellence on Complexity and Topology in Quantum Matter – ct.qmat Project-ID 390858490—EXC 2147. The work of Y. L. is funded through a PhD scholarship of the Studienstiftung des deutschen Volkes. D.R.F. is also supported by the Dutch Research Council (NWO) project 680-91-116 (Planckian Dissipation and Quantum Thermalisation: From Black Hole Answers to Strange Metal Questions), the FOM/NWO program 167 (Strange Metals) and through the grants SEV-2016-0597, PGC2018-095976-B-C21 and "María Zambrano para la atracción de talento" CA3/RSUE/2021-00898. Z. Y. X. also acknowledges support from the National Natural Science Foundation of China under Grants No. 11875053 and No. 12075298. Z. C. is also funded by China Scholarship Council.

# A    A fermion model with $\mathcal{PT}$ symmetry

In this appendix, we will give a simple model with $\mathcal{PT}$ symmetry, and to be specific, consider the $1 + 1$ dimensional Hamiltonian of fermion as in [91],

$$H = \int dx \left( -i\bar{\psi}\overleftrightarrow{\nabla}\psi + NO_1 \right) = \int dx \left( -i\bar{\psi}\overleftrightarrow{\nabla}\psi + NO^\dagger + NO \right), \tag{77}$$

$$O_1 = \bar{\psi}\psi, \quad O_5 = \bar{\psi}\gamma_5\psi, \quad O_1 = O^\dagger + O, \quad O_5 = O^\dagger - O, \tag{78}$$

with $N \in \mathbb{R}$ and $\bar{\psi} = \psi^\dagger\gamma_0$. In $1 + 1$ dimensional spacetime, the conventions are adopted as follows,

$$\gamma_0 = \sigma_1, \quad \gamma_1 = i\sigma_2, \quad \gamma_5 = \gamma_0\gamma_1 = \sigma_3, \tag{79}$$

where $\sigma_{1,2,3}$ are Pauli matrices. To relate it to the $\mathcal{PT}$ symmetry in the main text, one can consider a redefinition of operators as the last step of (77). The actions of $\mathcal{P}$ and $\mathcal{T}$ transformations on the fermion are

$$\mathcal{P}\psi(x,t)\mathcal{P} = \gamma_0\psi(-x,t), \quad \mathcal{P}\bar{\psi}(x,t)\mathcal{P} = \bar{\psi}(-x,t)\gamma_0, \tag{80}$$

$$\mathcal{T}\psi(x,t)\mathcal{T} = \gamma_0\psi(x,-t), \quad \mathcal{T}\bar{\psi}(x,t)\mathcal{T} = \bar{\psi}(x,-t)\gamma_0. \tag{81}$$

The transformations on the scalar and pseudo-scalar are the following,

$$\mathcal{P}O_1(x,t)\mathcal{P} = O_1(-x,t), \quad \mathcal{P}O_5(x,t)\mathcal{P} = -O_5(-x,t), \tag{82}$$

$$\mathcal{P}O(x,t)\mathcal{P} = O^\dagger(-x,t), \quad \mathcal{P}O^\dagger(x,t)\mathcal{P} = O(-x,t), \tag{83}$$

$$\mathcal{T}O_1(x,t)\mathcal{T} = O_1(x,-t), \quad \mathcal{T}O_5(x,t)\mathcal{T} = -O_5(x,-t), \tag{84}$$

$$\mathcal{T}O(x,t)\mathcal{T} = O^\dagger(x,-t), \quad \mathcal{T}O^\dagger(x,t)\mathcal{T} = O(x,-t). \tag{85}$$

So obviously, the Hamiltonian $H$ is Hermitian and also satisfies $\mathcal{P}$ and $\mathcal{T}$ symmetries, respectively

$$H = H^\dagger = \mathcal{P}H\mathcal{P} = \mathcal{T}H\mathcal{T}. \tag{86}$$

Furthermore, we consider the the charge transformation and chiral transformation

$$\psi \to e^{-i\varphi}\psi, \quad \psi \to e^{-i\theta'\gamma^5}\psi, \tag{87}$$

respectively. The theory is invariant under the charge transformation for general $N$ and is invariant under the chiral transformation when $N = 0$. The charge current and axis current corresponding to these symmetries are respectively

$$j^\mu = \bar{\psi}\gamma^\mu\psi, \quad j_5^\mu = \bar{\psi}\gamma^\mu\gamma_5\psi, \tag{88}$$

both of which are Hermitian. Although the chiral symmetry is broken when $N \neq 0$, the Ward identity still vanishes, i.e.

$$\langle\partial_\mu j_5^\mu\rangle = iN\langle O_5\rangle = iN\langle\mathcal{P}O_5\mathcal{P}\rangle = -iN\langle O_5\rangle = 0, \tag{89}$$

where $\langle X\rangle$ denotes $\text{Tr}[Xe^{-\beta H}]$. It shares the same feature as the holographic Ward identity (26).

To construct non-Hermitian $\mathcal{PT}$-symmetric Hamiltonian, one can define the Dyson map as the chiral transformation with the generator being the (rescaled) axis charge

$$Q = -\frac{1}{2}\int dx\,\psi^\dagger\gamma_5\psi. \tag{90}$$

So the Dyson map transforms the scalars as the following forms [91]

$$e^{\theta Q}Oe^{-\theta Q} = e^\theta O, \quad e^{\theta Q}O^\dagger e^{-\theta Q} = e^{-\theta}O^\dagger, \tag{91}$$

with $\theta \in \mathbb{C}$. Ultimately, the Dyson map transforms the Hamiltonian $H$ into a non-Hermitian one

$$\begin{aligned}
H_\theta &= e^{\theta Q}He^{-\theta Q} \\
&= \int dx\left(-i\bar{\psi}\slashed{\nabla}\psi + MO^\dagger + \bar{M}O\right) = \int dx\left(-i\bar{\psi}\slashed{\nabla}\psi + m_1O_1 + m_2O_5\right),
\end{aligned} \tag{92}$$

where

$$e^{-\theta}N = M = m_1 + m_2, \quad e^\theta N = \bar{M} = m_1 - m_2, \quad \tanh\theta = m_2/m_1. \tag{93}$$

Now we realize that the Dyson map corresponds to a complexified $U(1)$ transformation, which is analogous to (12). Consequently, both $M\bar{M} = N^2$ and $e^{\theta Q}O^\dagger Oe^{-\theta Q} = O^\dagger O$ are invariant under the complexified $U(1)$ transformation. The original theory has the non-negative invariant $N^2$. However, we could generalize it to the whole real axis $\mathbb{R}$. The source terms in $H_\theta$ share the same form as the holographic model (4) and rotor model (64).

One can check that both of the currents in (88) commute with $Q$. Thus, their transports are invariant under the Dyson map.

The $H_\theta$ with $M\bar{M} = N^2 \in \mathbb{R}$ is invariant under the combination of another $\mathcal{P}_\theta$ and $\mathcal{T}$ transformations. We define a new parity $\mathcal{P}_\theta$ as

$$\mathcal{P}_\theta = e^{2\text{Im}\theta Q}P. \tag{94}$$

$H_\theta$ is non-Hermitian but is $\mathcal{P}_\theta\mathcal{T}$-symmetric, namely

$$H_\theta^\dagger \neq H_\theta, \quad \mathcal{P}_\theta\mathcal{T}H_\theta\mathcal{P}_\theta\mathcal{T} = H_\theta, \tag{95}$$

where we have used $\mathcal{PT}Q\mathcal{PT} = Q$ [3].

# B   Instability analysis

We study the stability of the solution in holography by examining the QNM. Following [1], we only consider the perturbation on the matter fields, as this set will provide a consistent set of equations alone. The following ansatz of perturbations is introduced

$$\delta A = (a_t(z)dt + a_x(z)dx)e^{-i\omega t + ikx}, \quad \delta\phi = -\delta\bar{\phi} = \delta\varphi(z)e^{-i\omega t + ikx}, \tag{96}$$

so that the perturbation on energy-momentum tensor vanishes, that is $\delta T_{\mu\nu} \sim \phi\delta\bar{\phi} + \bar{\phi}\delta\phi = 0$. The perturbation equations are

$$kue^{-\chi}z^2 a_x' + \omega z^2 a_t' + 2qu\varphi e^{-\chi}\delta\varphi' - 2\delta\varphi que^{-\chi}\varphi' = 0, \tag{97}$$

$$\frac{ke^{\chi}\omega a_t}{u^2} + a_x\left(\frac{e^{\chi}\omega^2}{u^2} - \frac{2q^2\varphi^2}{uz^2}\right) + a_x'\left(\frac{v\varphi^4}{2uz} - \frac{\varphi^2}{uz} + \frac{3(u-1)}{uz}\right) + a_x'' + \frac{2\delta\varphi kq\varphi}{uz^2} = 0, \tag{98}$$

$$\frac{kq\varphi a_x}{u} + \frac{q\varphi e^{\chi}\omega a_t}{u^2} + \delta\varphi'' + \delta\varphi\left(\frac{u\left(\frac{2}{z^2} - k^2\right) + e^{\chi}\omega^2}{u^2} - \frac{2v\varphi^2}{uz^2}\right) +$$

$$\delta\varphi'\left(\frac{v\varphi^4}{2uz} - \frac{\varphi^2}{uz} + \frac{u-3}{uz}\right) = 0. \tag{99}$$

From applying local gauge transformation, we find that

$$a_t = -\omega, \quad a_x = k, \quad \delta\varphi = q\varphi(z) \tag{100}$$

is always a solution. To calculate the QNM on the asymptotic boundary, we require zero sources up to the gauge transformation (100), namely

$$qa_t(0)\varphi'(0) + \omega\delta\varphi'(0) = qa_x(0)\varphi'(0) - k\delta\varphi'(0) = 0. \tag{101}$$

Near the horizon, we impose ingoing boundary conditions and find the following expansions

$$a_t = (z_h - z)^{-i\omega/(4\pi T)+1}\left(-\frac{ka_{xh} + 2q\delta\varphi_h\varphi_h}{1/(4\pi T) + i/\omega}e^{-\chi_h/2} + O(z_h - z)\right), \tag{102}$$

$$a_x = (z_h - z)^{-i\omega/(4\pi T)}\left(a_{xh} + O(z_h - z)\right), \tag{103}$$

$$\delta\varphi = (z_h - z)^{-i\omega/(4\pi T)}\left(\delta\varphi_h + O(z_h - z)\right), \tag{104}$$

where the two coefficients $a_{xh}$ and $\delta\varphi_h$ are determined by the asymptotic boundary conditions (101).

Given a background solution, frequency $\omega$ and momentum $k$, we can transform the QNM of the perturbation equations (97) and the boundary conditions (101)(102) into the null vector of a matrix $\tilde{\mathcal{D}}$. By finding the poles of $1/\det\tilde{\mathcal{D}}$ in the complex $\omega$ plane with a given $k$, we can find the QNM. We show the most important pole in Fig. 12, where the frequency $\omega$ could cross the upper half plane at low momentum, which would signal the onset of the instability. However, we find that this mode triggers a shift on the chemical potential with $a_t(0) = \delta\mu$.

# C   Analytic conductivity in the probe limit

We aim here to derive an approximate expression of the conductivity and study the sum rule on the neutral background with small source. For this, we disregard the normalizable

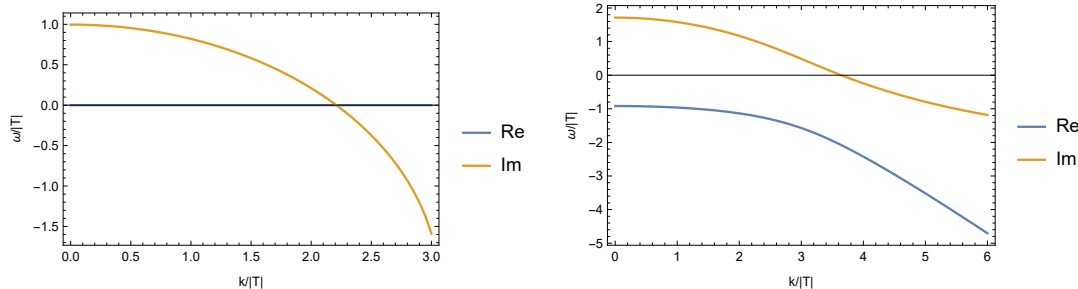

Figure 12: The unstable QNMs in phase II and III (from the left to the right) with $N/|T| = 1.71, 2.15$ respectively.

mode in the near boundary for the scalar field in the equation (59), so that $\varphi \approx Nz$ [2]. Adopting the notation $2q^2 N^2 \equiv k^2$ and rescaling $z_h \to 1$, we rewrite (59) as

$$a'' + \frac{u'}{u} a' + \left( \frac{\omega^2}{u^2} - \frac{k^2}{u} \right) = 0. \tag{105}$$

where $k^2$ is positive (negative) in phase I (phase II). The solution of this differential equation is expressed in terms of the Heun function $H_l(a, q; \alpha, \beta, \gamma, \delta; z)$

$$a(z) = c(1-z)^{\frac{-i\omega}{3}} (z-z_0)^{\frac{-i\omega}{3z_0^2}} (z-z_0^2)^{\frac{-i\omega}{3z_0}}$$
$$\cdot H_l \left( -z_0, -\frac{k^2}{z_0^2 - 1}; 0, 2, 1 - \frac{2i\omega}{3}, 1 - \frac{2i\omega}{3z_0}; \frac{1-z}{1-z_0^2} \right), \tag{106}$$

where $c$ is an integration constant and $z_0 = \frac{-1+\sqrt{3}i}{2}$. The retarded Green's function and the conductivity are found to be

$$G_R(\omega) = \frac{1}{3(z_0^2 - 1)}$$
$$\cdot \left( \frac{3H_l' \left( -z_0, -\frac{k^2}{z_0^2-1}; 0, 2, 1 - \frac{2i\omega}{3}, 1 - \frac{2i\omega}{3z_0}; \frac{1}{1-z_0^2} \right)}{H_l \left( -z_0, -\frac{k^2}{z_0^2-1}; 0, 2, 1 - \frac{2i\omega}{3}, 1 - \frac{2i\omega}{3z_0}; \frac{1}{1-z_0^2} \right)} + \frac{i\omega(z_0^5 - z_0^3 + 2z_0^2 - 2)}{z_0^3} \right), \tag{107}$$

and

$$\sigma(\omega) = 1 + \frac{1}{\sqrt{3} - 3i} \left( \frac{2H_l' \left( \frac{1-i\sqrt{3}}{2}, -\frac{2ik^2}{\sqrt{3}-3i}; 0, 2, 1 - \frac{2i\omega}{3}, 1 - \frac{4\omega}{3\sqrt{3}+3i}; \frac{-2i}{\sqrt{3}-3i} \right)}{\omega H_l \left( \frac{1-i\sqrt{3}}{2}, -\frac{2ik^2}{\sqrt{3}-3i}; 0, 2, 1 - \frac{2i\omega}{3}, 1 - \frac{4\omega}{3\sqrt{3}+3i}; \frac{-2i}{\sqrt{3}-3i} \right)} \right), \tag{108}$$

where the prime in the numerator denotes the $z$-derivative. In the low frequency limit, the conductivity takes the form

$$\sigma(\omega) \approx 2N^2 q^2 \left( \pi \delta(\omega) + \frac{i}{\omega} \right) + \text{regular terms.} \tag{109}$$

This result is compatible with (62) at $T \to 0$ (notice that this implies $\rho_n \to 0$), and the leading term of $\rho_s$ takes $2N^2$.

The limit we take in this section is equivalent to $|k| \ll 1$, within this constraint we found out that the sum rule still holds. As can be seen from Fig. (13), in the high frequency regime, the integrated spectral weight (55) decreases in power law (left) and there is no pole in the upper-half plane and the real axis of the complex frequency (right). This justifies the two conditions from [55].

---

[2] We thank Jie Ren for helpful discussions and comments.

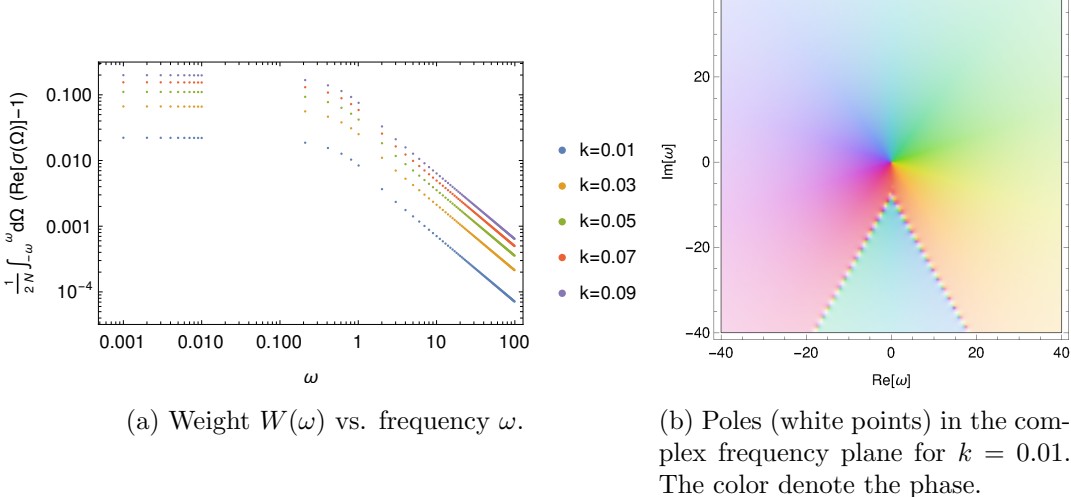

(a) Weight $W(\omega)$ vs. frequency $\omega$.

(b) Poles (white points) in the complex frequency plane for $k = 0.01$. The color denote the phase.

Figure 13: (a) Integral weight $S_\sigma(\Omega)/N$ as a function of the frequency. From the scaling-law decay at large $\omega$, the second condition of the sum rule is proved. (b) No poles on the upper-half plane and the real axis of complex frequency, this proves the first condition of the sum rule. Both figures correspond to phase I.

## D   Complex temperature of the double cone geometry

Complex metrics and complex time periods are not rare in holography. We take the double cone geometry in [17] as an example, where the bulk theory is JT gravity and the boundary theory is a $0 + 1$ Schwarzian action with a Hermitian Hamiltonian $H$. The double cone geometry is dual to the ramp of the spectral form factor (SFF) of the boundary theory

$$\mathrm{Tr}[e^{-(\beta+iT)H}]\mathrm{Tr}[e^{-(\beta-iT)H}], \quad \beta, T \in \mathbb{R}, \tag{110}$$

which has complex inverse temperatures $\beta + iT$ and $\beta - iT$. The ramp in the connected part of SFF is contributed by a semiclassical double cone geometry, with metric

$$ds^2 = -\left(\sinh\rho + i\frac{\beta}{T}\cosh\rho\right)^2 d\tilde{t}^2 + d\rho^2, \quad \tilde{t} \sim \tilde{t} + \tilde{T}, \quad \tilde{T} \in \mathbb{R}, \tag{111}$$

where the periodicity $\tilde{T}$ defined in [17], is the auxiliary inverse temperature parameterizing the solutions that contribute to the ramp. For the right factor $\mathrm{Tr}[e^{-(\beta-iT)H}]$ in (110), the induced metric on the right UV cutoff slice of the double cone geometry satisfies the boundary condition

$$\frac{d\tau_{\mathrm{bdy}}^2}{\epsilon^2} = -\left[\frac{e^{\rho_\epsilon}}{2}\left(1 + i\frac{\beta}{T}\right)\right]^2 d\tilde{t}^2. \tag{112}$$

Upon identification of $T = \frac{1}{2}\epsilon e^{\rho_\epsilon}\tilde{T}$, this agrees with a complex period in $\tau_{\mathrm{bdy}} \sim \tau_{\mathrm{bdy}} + \beta - iT$, where a possible real factor was absorbed into $\tilde{T}$. In summary, both of the line elements in (41) and (112) are complex, and hence the coordinates $\tau$ in (41) and $\tilde{t}$ in (112) play the similar roles.

# E   A two-level system with $\mathcal{PT}$ symmetry

Here we review the non-Hermitian two-level model with $\mathcal{PT}$ symmetry and discuss the $\mathcal{PT}$ symmetry breaking at finite temperature. Consider a non-Hermitian Hamiltonian of two-level system

$$H = \sigma_1 + i\sqrt{1 - N^2}\sigma_3, \tag{113}$$

where $N^2 \in \mathbb{R}$ and $\sigma_{1,3}$ are Pauli matrices. It is $\mathcal{PT}$-symmetric with $\mathcal{P} = \sigma_1$ being parity and $\mathcal{T}$ being the complex conjugate. The eigenvalues are $E_\pm = \pm N$. They are real when $N^2 \geq 0$ and imaginary when $N^2 < 0$. So the $\mathcal{PT}$ symmetry is spontaneously broken at zero temperature when $N^2 < 0$.

Consider the canonical ensemble $e^{-\beta H}$ at finite temperature $T = 1/\beta$. The partition function, free energy, energy, and average entropy are respectively

$$Z = 2\cosh(\beta N), \tag{114}$$

$$F = -\frac{1}{\beta}\log(2\cosh(\beta N)), \tag{115}$$

$$E = -N\tanh(\beta N), \tag{116}$$

$$S = \log(2\cosh(\beta N)) - \beta N\tanh(\beta N). \tag{117}$$

We plot $\text{Re}F$ on the $N^2$-$T$ plane in Fig. 14. We call the region of $N^2 \geq 0$ phase I, in which the spectrum and the free energy are both real. The region of $N^2 < 0$ and $|N| < \pi T/2$ is phase II, in which the spectrum is complex but the free energy remains real. The region of $N^2 < 0$ and $|N| > \pi T/2$ is phase III, in which the spectrum is complex and the free energy encounters the first branch cut along the real axis of $\beta$ at $\beta N = \pi/2$, as shown in Fig. 15. $\beta$ must deviate from the real axis and continue to the upper or lower half plane such that $F$ takes complex conjugate values on the two half planes. The phase structure presented in Fig. 14 is analogous to that in the holographic model as shown in Fig. 1. The emergence of complex temperature on the phase boundary between phases II and III also evokes the complex solutions with complex temperatures, as we find in holography.

Furthermore, in order to get the eigenvalues $E_\pm$ from the canonical ensemble in the zero-temperature limit, we could introduce an imaginary angle to the inverse temperature $\beta = |\beta|(1 \pm i\epsilon)$ with $\epsilon > 0$ and send $|\beta| \to \infty$. Taking the branch from analytical continuation, we get the two eigenvalues from both free energy $F$ and average energy $E$,

$$F \to \pm N, \quad E \to \pm N. \tag{118}$$

Similar limit behavior appears in the holographic model where the $N^2/|T|^2 \to -\infty$ limit of the solutions in phase III gives the zero temperature solutions.

# F   Holographic renormalization

By using the holographic renormalization in [92,93], we can determine the thermodynamic quantities including the grand potential density $\omega_G$, charge density $\rho$, energy density $\varepsilon$, and pressure $p$. We follow the same approach as in [94, 95]. Firstly, in order to derive the on-shell action $S_{\text{on-shell}}$, we need the extrinsic curvature $K_{\mu\nu}$ and scalar $K$, which are defined by

$$K_{\mu\nu} = \frac{1}{2\sqrt{g_{zz}}}\partial_z\gamma_{\mu\nu}, \quad K = K_{\mu\nu}\gamma^{\mu\nu}, \tag{119}$$

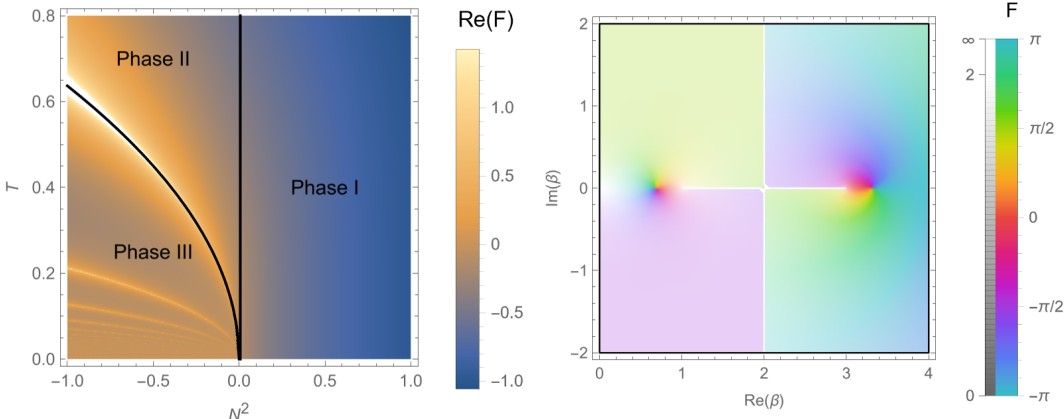

Figure 14: The real part of free energy $\mathrm{Re}(F)$ on the $N^2$-$T$ plane. The black curves denote the phase boundaries.

Figure 15: Free energy $F$ on the complex plane of $\beta$ at $N = i\pi/2$. The white segments are branch cuts. The argument $\arg(F)$ is denoted by color.

where $\gamma_{\mu\nu}$ is the induced metric on a constant $z$ slice, where $\mu, \nu$ stand for the $(t, \mathbf{x})$ indexes. Inserting the on-shell relation for the Ricci scalar,

$$R = 2V + D_a^\dagger \bar\phi D^a \phi, \qquad V = -\frac{d(d-1)}{L^2} + m^2 \bar\phi\phi - v\bar\phi^2\phi^2 \,, \tag{120}$$

into the action (4) and after some further manipulations, we arrive at[3]

$$S_{\text{on-shell}} = \int_{z_\Lambda}^{z_h} d^4x \partial_z \left[ -\frac{2}{3}\left(\sqrt{-\gamma}K\right) - \frac{1}{3}A\left(\sqrt{-g}F^{z0}\right) \right] + 2\int_{z_\Lambda} d^3x \sqrt{-\gamma}K \,, \tag{121}$$

where the first term is split into two surface integrals on the boundary and horizon from Gauss theorem. The action (121) diverges when $z_\Lambda \to 0$. This can be canceled by adding the following boundary counterterms (we set $\Delta = 2$)

$$S_{\text{ct}} = \int_{z=z_\Lambda} d^3x \sqrt{-\gamma}\left[4 + \phi\bar\phi\right] \,, \tag{122}$$

so that the renormalized action $S_{\text{ren}}$ in (7) reads

$$S_{\text{ren}} = \lim_{z_\Lambda \to 0}\left[S_{\text{on-shell}} + S_{\text{ct}}\right] \,. \tag{123}$$

Near the boundary $(z \to 0)$, the bulk fields read

$$\phi = Mz + \phi_2 z^2 + \cdots, \quad \bar\phi = \bar{M}z + \bar\phi_2 z^2 + \cdots,$$
$$u = 1 + \frac{M\bar{M}}{2}z^2 + u_3 z^3 + \cdots, \quad \chi = \frac{M\bar{M}}{2}z^2 + \frac{2}{3}(M\bar\phi_2 + \bar{M}\phi_2)z^3 + \cdots, \tag{124}$$
$$A = \mu + a_1 z + \cdots.$$

To express the horizon integral in (121) in terms of the boundary data, we evaluate the constraint equation (30) on the boundary and horizon, namely

$$\frac{1}{4\pi}\left[-\mu\rho + 2(M\bar\phi_2 + \bar{M}\phi_2) - 3u_3\right] = \frac{Ts}{4\pi} \,, \tag{125}$$

---

[3]In this section, as well as in the whole work, we employ $G_N = \frac{1}{16\pi}$ units.

where $T$ is the black hole temperature and $s$ is the entropy density. Finally, we express the renormalized action (123) in terms of the boundary data, obtaining

$$\omega_G = u_3 - (M\bar{\phi}_2 + \bar{M}\phi_2)\,, \tag{126}$$

The remaining thermodynamic quantities can be determined by considering variations of the of the renormalized action with respect to the boundary sources

$$\langle T^{\mu\nu}\rangle = -2\lim_{z\to 0}\frac{1}{z^2}\left[-\sqrt{-\gamma}\Pi^{\mu\nu} + \frac{\delta S_{ct}}{\delta\gamma_{\mu\nu}}\right]\,, \tag{127}$$

$$\langle\mathcal{O}\rangle = -\lim_{z\to 0}z\left[-\sqrt{-g}g^{zz}\partial_z\phi + \frac{\delta S_{ct}}{\delta\bar{\phi}}\right]\,, \tag{128}$$

$$\langle\mathcal{O}^\dagger\rangle = -\lim_{z\to 0}z\left[-\sqrt{-g}g^{zz}\partial_z\bar{\phi} + \frac{\delta S_{ct}}{\delta\phi}\right]\,, \tag{129}$$

$$\langle J^\mu\rangle = -\lim_{z\to 0}\left[\sqrt{-g}g^{zz}g^{\mu\alpha}F_{z\alpha}\right]\,, \tag{130}$$

where $\Pi^{\mu\nu} = K^{\mu\nu} - \gamma^{\mu\nu}K$ is the Brown-York tensor. Evaluating the expressions (127)-(130) at the boundary by means of the expansions (124) yields

$$\langle T^{00}\rangle = \varepsilon = -2u_3 + M\bar{\phi}_2 + \bar{M}\phi_2\,, \tag{131}$$

$$\langle T^{ii}\rangle = p = -u_3 + M\bar{\phi}_2 + \bar{M}\phi_2\,, \tag{132}$$

$$\langle\mathcal{O}\rangle = \phi_2\,, \tag{133}$$

$$\langle\mathcal{O}^\dagger\rangle = \bar{\phi}_2\,, \tag{134}$$

$$\langle J^0\rangle = \rho = -a_1\,, \tag{135}$$

From here, we notice that the pressure $p = -\omega_G$, as expected. In addition, the Ward identity for the trace of the stress tensor reads

$$\langle T^\mu_{\ \mu}\rangle = \eta_{\mu\nu}\langle T^{\mu\nu}\rangle = M\left\langle\mathcal{O}^\dagger\right\rangle + \bar{M}\left\langle\mathcal{O}\right\rangle\,, \tag{136}$$

which has the expected form. Furthermore, after combining (131) and (132), we find the Gibbs-Duhem relation

$$\varepsilon + p = -3u_3 + 2(M\bar{\phi}_2 + \bar{M}\phi_2) = \mu\rho + Ts\,, \tag{137}$$

where we have made use of the constraint (125), as well as of (135).

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
