# Peer review of "Electric conductivity in non-Hermitian holography"

_SciPost Physics_

## Round 1 · Referee Report · Anonymous (Referee 1) · 2023-7-13

Strengths

1- The paper is well written 2- The presentation is clear and accessible to a wide audience 3- The results are original 4- Although is strongly based on ref [1] the treatment is self-contained, to the point that there is no need to revisit [1] to go through this work. 5- It gives a complementary approach to the holographic studies based on a weakly interacting model

Weaknesses

1- As written it is not completely clear the relevance of studying non-hermitian systems at finite chemical potential 2- The weakly interacting model presented in section 4 does not present a normal contribution to the conductivity. This makes the connection to the holographic model less evident. This is supposed to be fixed by a one loop computation.

Report

In my opinion, the strengths of the paper widely surpass its weaknesses and it should be published in some form. But before that, there are some minor specific questions I would like to ask to the authors. I will list them in the section of requested changes although some of them might not merit a change in the draft.

Requested changes

1- The authors present the phase diagram for their holographic model at Fig 1. The presentation is catchy but has the an inconvenient. It is hard to imagine what does $N^2$ mean at that stage of the paper. It is only defined several pages latter in Eq (13). The authors should refer to this equation and give at least an informal definition of $N^2$, explaining at least what does the sign of $N^2$ stands for. In this way potential readers can profit from the presentation.

2- In Section 2.3.1 the authors mention the c-theorem. I guess they read c from the effective AdS radius in the IR. Is that correct? They also mention that it satisfies a c-theorem. Can the authors recall the ingredients for a holographic c-theorem? Do all proofs rely on NEC? Maybe a reference (Myers-Sinha?) and a clarification on why the c-theorem still works will be helpful.

3- In the description of phase I in page 10. Is there an interpretation for the sign of $\langle O \rangle$?

4- In the description of phase II in page 11 the authors claim "these two branches are both unstable in the sector of scalar $(A_x, \phi)$ perturbations". I think there is a typo there. At $k=0$ it should be $A_t$ instead of $A_x$. This is consistent with the computations of appendix B. An instability on the $A_x$ sector would imply a non-analyticity in the upper complex plane of $G_{J_xJ_x}$, which would imply a violation of the sum rule for the conductivity.

5- Is there a hint of an end point for this unstable backgrounds? Or a reason to imagine that there will be no static end point to this instability?

6- For solutions with complex temperatures, is there a physical reason for presenting the solutions in terms of $|T|$, or it is just for presentation? Also is some of the phase diagrams the axe is labeled by $T$, while I imagine it should be $|T|$ to make sense with phase III solutions. See for instance Fig 1, Fig 14.

7- In section 2.4. What does it mean the superconducting phase transition in backgrounds where the $U(1)$ symmetry is explicitly broken? Is it just a second branch of solutions that continuously connects to the HHH when both $M$ parameters are turned off? Is it always the stable branch? what does "cross-over" stands for? Is the free energy and all its derivatives continuous?

8- About section 3. How does your results (if they do at all) connect with 2104.02428? In particular, the authors of 2104.02428 claim that a) The sum rule is satisfied even in tachionic backgrounds b) They blame for that the local conservation of the current, much alike your Ward identity (26) when evaluated in a static background. c) They give a connection between $\sigma_Q$, the imaginary mass and the effective gap

9- What is the explicit expression for ${\cal M}$ in (71)?

10- The free model of Section 4 could easily incorporate a chemical potential. Is there a reason why the authors did not consider that? Will its effect be a simple shift of the mass parameter $r$?

11- Finally, I understand that studying interacting systems away from charge neutrality is one of the biggest strengths of AdS/CMT, as other methods usually fail. But I feel that the motivation should be more explicit. What is the state of art in the context of non-hermitian physics? Or was the general idea of turning on $\mu$ to give the sum rule another chance to fail?

  • validity: good
  • significance: good
  • originality: good
  • clarity: good
  • formatting: good
  • grammar: good

Author:  David Rodriguez  on 2023-11-08  [id 4101]

(in reply to Report 1 on 2023-07-13)
Category:
answer to question

We thank the referee for the careful consideration and valuable comments. In the pdf file attached, we address to each one of them, and indicate the changes made to the manuscript accordingly.

Attachment:

ECNHH_reply_to_referee_1.pdf

---

## Round 1 · Referee Report · Anonymous (Referee 2) · 2023-8-1

Strengths

1- The paper is well written, and the presentation is clear and reasonably self-contained. 2- As discussed in the introduction, the study of PT-symmetric non-Hermitian systems is important, particularly in the context of condensed matter theory. Holography is a valuable tool when considering strongly coupled models in this context. The relevance of extending the model of Ref[1] by including a non zero chemical potential and studying the corresponding transport properties, which are the main goals of this manuscript, is then clearly motivated. 3- The results are original and interesting, particularly so for the pattern of enhancement/suppression of the quantum critical conductivity in the different phases, and also the fact that the FGT sum rule is always satisfied.

Weaknesses

1- While the motivations of the authors are clear, I believe the paper lacks a detailed presentation of the state-of-the-art regarding the main ingredients of the model and the results of this manuscript. I address this more precisely in the "Requested changes".

2- The authors claim that "complex metrics and temperatures are not rare in holography", but then only discuss one particular instance with these features, the so-called double cone geometry, very briefly and in appendix C. This seems a bit too quick to put the results of this model in context.

3- The structure of the paper is slighly surprising: why is the bosonic toy model described in Sec.4 in the main text while the fermionic toy model is relegated to Appendix A? If the role of these toy models is simply to have some comparison with the holographic computations, I would think that this is less relevant than the actual analytic computation in the probe limit, which is relegated to Appendix B.

Report

This paper extends the model of Ref.[1] to include the chemical potential and study electric conductivity of a PT-symmetric non-Hermitian holographic model.

I believe the paper is well written and contains interesting results. Consequently, in my opinion it should be published eventually. However, the authors should first address the comments in the "Weaknesses" and "Requested changes sections".

Requested changes

As stated above, I the present manuscript should include a more detailed description of the state-of-the-art and a discussion of their results in the context of previous expectations:

1) What is known about electric conductivity and chemical potentials in PT-symmetric systems?

2) How is this affected by weak coupling vs strong coupling?

3) How do the results presented here compare with previous expectations? For instance, was it expected that the FGT sum rule should actually break down?

4) What is the relation between the appearence of complex VEVs, metrics and temperatures in phase III of the model as compared to other instances where these features have appeared in holography? For instance, is there a connection with the so-called complex CFTs?

  • validity: good
  • significance: good
  • originality: good
  • clarity: high
  • formatting: good
  • grammar: excellent

Author:  David Rodriguez  on 2023-11-08  [id 4100]

(in reply to Report 2 on 2023-08-01)
Category:
answer to question

We thank the referee for the valuable comments in relation to our work. Below we address to each of the requested changes. The citations referred here are those linked to the main draft (in the bibliography), which we resubmitted, after modifications.

Q: What is known about electric conductivity and chemical potentials in PT-symmetric systems?

A: To the best of our knowledge, not many results exist on electric transport in non-interacting, and particularly even less in interacting PT-symmetric systems. In [40], the authors study the conductivity in a 1-dimensional non-Hermitian Dirac model, which has a PT-symmetric phase and a PT-broken phase at zero temperature. They find that the sum rule holds in both phases, as a consequence of the U (1) gauge invariance. In [83], a PT-symmetric superconductor junction was considered, where the U(1) symmetry is spontaneously broken and the current-voltage characteristic strongly depends on the chosen inner product. For the model in our paper, the U (1) symmetry is explicitly broken by the non-Hermitian source deformations, which constitutes a physically different situation from [40,82,83].

Q: How is this affected by weak coupling vs strong coupling?

A: Other than from our work, to date no results for electric transport in PT symmetric non-Hermitian systems exists in the strongly interacting regime. In our holographic model, both the UV and IR fixed points are conformal. The holographic system is gapless, and also does not have a quasi-particle description. The UV fixed point leads to the limiting behavior σ(ω) → 1 at high frequency. In addition, at zero temperature, the conductivity does not have a hard, but rather a soft gap, which arises in such holographic models [73]. This is to be contrasted to the rotor model, where the pseudo-Goldstone mode α in (69) and the fluctuation (71) are particle excitations. On the other hand, the model of [40] has fermionic quasiparticle excitations, and in the PT-symmetric phase, the fermion has a finite mass and the conductivity has a hard gap at half-filling. These differences between the holographic model and models of quasiparticle transport are qualitatively the same as in Hermitian systems, at least for weak PT deformation in phase I. Furthermore, our results for the conductivity in the strong PT deformation regime in phase II and III are completely novel.

Q: How do the results presented here compare with previous expectations? For instance, was it expected that the FGT sum rule should actually break down?

A: There are several reasons that could lead to an expectation of the sum rule not holding in our holographic model: 1) [56] showed that stability in the vector channel is necessary for the FGT sum rule to hold. On the other hand, [1] already showed that there is an unstable mode in the scalar (δAt, δφ) channel. Based on this, it was not clear to us that no instability should be present in the
vector channel either. We performed a nontrivial check by showing that such an 5 instability is absent, and hence the FGT sum rule actually holds. 2) Usually, it is expected that charge conservation leads to the sum rule [40]. There two reasons to believe that this argument may not hold in our model: First, the instability in the scalar channel involves the charge density mode δAt, and hence evolving this instability might violate charge conservation. Second, we already argue in the reply to question 8 of Referee 1 that the charge conservation equation in our model does not hold as an operator identity, but only in the static state. Hence, an independent check of the FGT sum rule also seemed necessary from the point of view of charge conservation. Our preliminary conclusion is that the implication of [Phys. Rev. B 71, 104511 (2005)] might rely on too strong assumptions, and the charge conservation in the considered state (not in all states) may be enough to show the FGT sum rule.

Q: What is the relation between the appearance of complex VEVs, metrics and temperatures in phase III of the model as compared to other instances where these features have appeared in holography? For instance, is there a connection with the so-called complex CFTs?

A: Complex CFT is one kind of non-unitary CFT. It appears when the Breitenlohner-Freedman bounds for some operators are broken and a pair of complex fixed points appears with the scaling dimensions being complex conjugate pairs [Phys. Rev. D 80, 125005]. However, in our PT symmetric system, the sources are complexified, while the scaling dimension of the scalar operators stays real in both the UV and IR fixed points, i.e. we are dealing with real CFTs. We hence do not see a direct connection with complex CFTs.

Attachment:

ECNHH_reply_to_referee_2.pdf

---

## Editorial Decision

resubmitted